# Efficient, biosafe and tissue adhesive hemostatic cotton gauze with controlled balance of hydrophilicity and hydrophobicity

Huaying He[1,5], Weikang Zhou[1,4,5], Jing Gao[1], Fan Wang[2], Shaobing Wang[2], Yan Fang[1✉], Yang Gao[1], Wei Chen [1,3✉], Wen Zhang[1], Yunxiang Weng[1], Zhengchao Wang [2] & Haiqing Liu [1,3✉]

Cotton gauze is a widely used topical hemostatic material for bleeding control, but its high blood absorption capacity tends to cause extra blood loss. Therefore, development of rapid hemostatic cotton gauze with less blood loss is of great significance. Here, we develop an efficient hemostatic cotton gauze whose surface is slightly modified with a catechol compound which features a flexible long hydrophobic alkyl chain terminated with a catechol group. Its hemostatic performance in animal injuries is superior to standard cotton gauze and Combat Gauze[TM]. Its biosafety is similar to cotton gauze and rebleeding hardly occurs when the gauze is removed. Here, we show its hemostatic capability is attributable to the rapid formation of big and thick primary erythrocyte clots, due to its effective controlling of blood movement through blocking effect from tissue adhesion by catechol, blood wicking in cotton, and the hydrophobic effect from long alkyl chains.

[1] College of Chemistry and Materials Science, Fujian Provincial Key Laboratory of Polymer Materials, Fujian Normal University, 350007 Fuzhou, China. [2] Provincial University Key Laboratory of Sport and Health Science, College of Life Sciences, Fujian Normal University, 350007 Fuzhou, China. [3] Engineering Research Center of Industrial Biocatalysis, Fujian Province Higher Education Institutes, 350007 Fuzhou, China. [4] Present address: Zijin Middle School, 364214 Shanghang, China. [5] These authors contributed equally: Huaying He, Weikang Zhou. ✉email: fangyan_YWJ@163.com; chenwei@fjnu.edu.cn; haiqingliu@fjnu.edu.cn

Massive bleeding from nonfatal traumatic wounds is one of the leading causes of death and disability in battlefields and civilian accidents, because significant blood loss causes symptoms such as hypothermia, coagulopathy, acidosis, sepsis, and organ failure[1,2]. More than 50% of these mortalities is preventable if emergent and efficient hemostatic measures are applied. This inspires the development of advanced hemostatic products and technologies for prehospital bleeding control of wounded people, in order to increase survival rate and reduce medical costs[3–6].

Cotton gauze has a long history as an effective topical hemostatic fabric for compressible and noncompressible wounds, mainly due to its safety, non-allergy, low cost, adaptability, breathability, stability, blood absorbency, and easy applicability[7,8]. It is still the most widely used hemostat for traumatic bleeding control, although lots of efficient hemostatic agents have been manufactured and clinically applied in recent two decades[9–12]. The hemostatic mechanism of cotton gauze counts on the activation of platelets upon contacting with cotton fiber, and its quick wicking of blood fluid, leading to resting of blood cells and platelets to form blood clots. However, it's often seen that excessive large volume of blood is absorbed by cotton gauze before bleeding stops, due to its highly hydrophilic nature, porous structure, and capillary action among the weaved fibers. Those extra blood losses may be the last straw that causes morbidity or mortality because the blood volume in the circulating system is critical.

Many endeavors to enhance the hemostatic efficacy (reduction in blood loss and bleeding time) of cotton gauze have been made in the academic and industrial circles. Z-Medica in USA has commercialized a gauze brand-named QuikClot Combat Gauze® (QCG), which is made by binding inorganic mineral kaolin particles onto rayon/polyester nonwoven. Kaolin can activate clotting factor XII to accelerate blood coagulation reactions, leading to fast thrombus formation. This topical hemostat has been clinically adopted in military, emergency care, and hospital for compressible severe hemorrhage. However, its rapid hemostasis efficacy is subsided because of possible loss of kaolin, and the detached kaolin particles may cause risks of unexpected distal thrombus. To increase binding stability of inorganic particles to

cotton fiber, mesoporous chabazite zeolite particles were chemically anchored onto fiber surface by an on-site growth route[8]. Such a composite cotton gauze has a better topical hemostatic efficiency than QCG. In the rabbit lethal femoral artery injury model, blood loss of chabazite zeolite-cotton gauze was only about 40% less than that of QCG. For these two kinds of hybrid cotton gauzes, it's obvious that they still absorb large precious volume of blood during bleeding control. Tuning the wettability of a hemostatic fabric was proposed to address this concern[13]. A Janus gauze consisting of a top hydrophobic fabric layer and a bottom hydrophilic cotton fabric layer was developed. It was thought that the hydrophilic layer absorbed blood to expedite clotting, while the hydrophobic layer gives a pressure to inhibit blood diffusion through gauze in lengthways. However, blood permeation in warp and weft directions of the bottom cotton fabric layer is un-avoidable, resulting in losses of valuable blood yet. Furthermore, in most cases, blood seepage from the seam between the hemostatic fabric/wound surface, which leads to massive blood loss as well. A composite gauze with a superhydrophobic Poly(vinylidene fluoride)/carbon nanofiber (PVDF/CNF) bottom coating layer was reported to have fast hemostatic capability and no re-bleeding potential, because of synergetic effects from CNF's acceleration of fibrin fiber formation and PVDF's repellency of blood[14]. Therefore, controlling of movement of blood fluid at the gauze/tissue contact surface and in gauze is the key in designing a highly efficient hemostatic gauze. Inspired by mussel foot protein's good adhesion to wet substrates[15–17], we speculate that cotton gauze with catechol group on its surface may help it adhere to blood-wetted tissue, in order to hinder blood seepage from the seam of gauze/tissue contact surface; meanwhile, the strong interaction among cotton fibers through catechol linkage may slow down blood diffusion in gauze.

In this work, in view of the great importance of fiber structure, wettability, and wet biological tissue adherence to hemostatic materials, we design and prepare a hemostatic cotton fabric, which is made by slightly grafting a catechol compound 1,2-benzenediol-3-(7,9,13-pentadecatrienyl) (USO, Fig. 1a) with a C15 alkyl side chain onto the fabric surface. This hemostatic

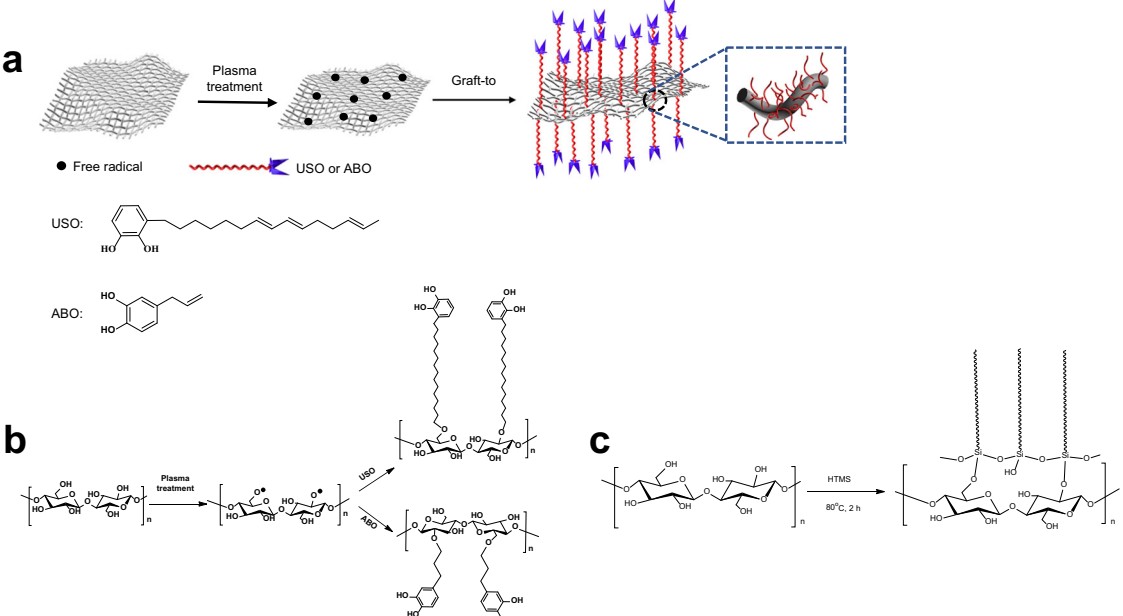

**Fig. 1 Schematic fabrication of surface modified cotton gauze. a** Reaction scheme for grafting catechols onto cotton gauze. **b** Surface grafting of ABO and USO onto cotton cellulose. **c** Surface grafting HTMS onto cotton cellulose.

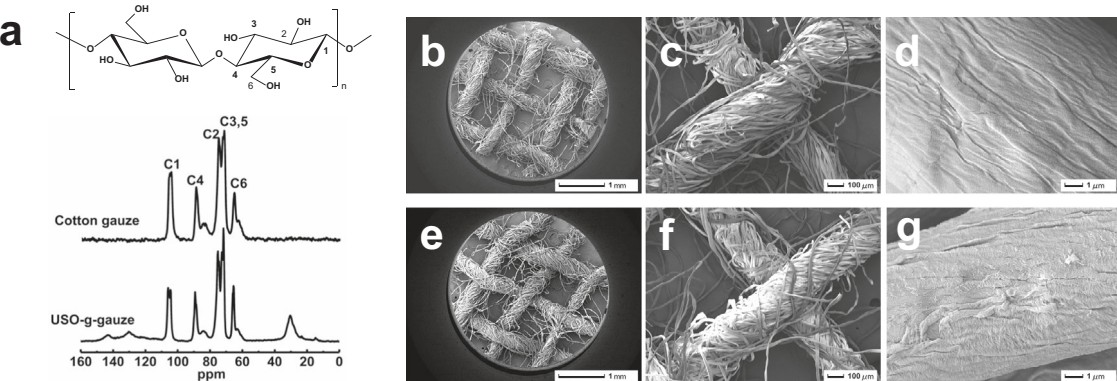

**Fig. 2 Chemical structure and morphology of gauzes. a** Solid-state $^{13}$C NMR spectra of gauzes. SEM images: **b–d** cotton gauze, and **e–g** USO-g-gauze at different magnifications. **b–g** Five spots were observed independently with similar results. Source data for **a** are provided as a Source Data file.

fabric integrates wet tissue adhesiveness from catechol group, hydrophobicity from long alkyl chain, absorbency and breathability from cotton fiber, into one hemostatic device. In rat femoral artery and liver laceration, and pig femoral artery injury models, this modified cotton gauze shows very limited blood permeation through the gauze, no blood oozing out from the seam of gauze/wound contact surface, short bleeding time and reduced blood loss. Its biosafety is also comparable to cotton gauze. This idea and methodology are also successfully applied to chitosan nonwoven fabric to fabricate a gauze with high hemostatic capability.

## Results and discussion

**Chemical composition and surface structure of gauze**. Free radicals can be readily generated on cotton gauze by plasma treatment. Through radical initiated reaction with the double bonds in USO, which can be grafted onto cotton gauze surface. Thus, a long alky side chain with a catechol end group was introduced onto cotton fiber (Fig. 1a, b). Its chemical structure is confirmed by its solid state $^{13}$C NMR spectrum (Fig. 2a). The peaks at region of 60–70 ppm are attributed to C6 of cellulose, while the signals at region of 70–80 ppm are assigned to C2, C3, and C5 of cellulose. The peaks at region of 80–95 and 100–110 ppm correspond to C4 and C1, respectively[18]. Compared with cotton gauze, several new peaks appear for USO-g-gauze, i.e., broad/weak peaks at 120–150 ppm corresponding to unsaturated carbons on benzene ring; peaks at region of 10–40 ppm assigning to methyl and methylene groups of the alkyl chain of USO. Therefore, $^{13}$C-NMR spectra indicates that USO is successfully grafted onto cotton gauze surface.

FTIR and XPS also prove that 4-allyl-1,2-benzenediol (ABO), hexadecyltrimethoxysilane (HTMS) and USO were successfully grafted onto the surface of cotton gauze (Supplementary Figs. 3 and 4). For cotton gauze, characteristic absorption bands at 3200–3600 cm$^{-1}$($\upsilon_{O-H}$), 1610 cm$^{-1}$($\upsilon_{C-C}$), 1435 cm$^{-1}$($\delta_{C-O-C}$), and 1125 cm$^{-1}$($\upsilon_{C-O-C}$) are present. In the spectra of ABO-g-gauze, HTMS-g-gauze, and USO-g-gauze, a new band at 2846 cm$^{-1}$ appears for $\upsilon_{C-H}$ of –CH$_2$– groups of ABO, HTMS, and USO grafted on cotton gauze (Supplementary Fig. 3). C$_{1s}$ XPS spectra shows the relatively strongest peak of USO-g-gauze is C–C bond, while that of cotton gauze is C–O–/C–OH bond (Supplementary Fig. 4b). This is in accordance with the abundance of C–C bond from the long aliphatic chain grafted on USO-g-gauze.

**Surface morphology of gauzes**. Cotton gauze consists of interwoven cotton yarns (Fig. 2b–g). Pores including macropores (among yarns), capillaries (among fibers), mesopores and micropores (inner fiber) are abundant in the gauze (Fig. 2b). After plasma treatment and radical initiated graft reaction with USO, the interwoven fiber network structure is well maintained, while the surface roughness of cotton fiber slightly increases (Fig. 2g). On the one hand, the improved surface roughness is due to the deepening effect of plasma etching[19]. On the other hand, the USO thin layer grafted on the fiber surface may also enhance roughness[20]. In fact, the well maintenance of fiber/yarn/fabric morphology is also found for ABO-g-gauze and HTMS-g-gauze (Supplementary Fig. 5).

**Wettability of gauzes**. The wettability of gauze has a large effect on blood fluid absorption, protein adsorption, and blood cell adhesion. For instance, it is reported that a material with a water contact angle (WCA) of 40–70° is suitable for adhesion of various cells[21]. The wettability of gauze by simulated body fluid (SBF) and fresh rat blood was evaluated by applying 200 μL of these liquids onto gauze. As shown in Fig. 3a, a water droplet immediately spreads over and diffuses into gauze upon dripping onto gauze surface, due to cotton gauze has a robust intrinsic hydrophilicity and capillary structure. ABO-g-gauze exhibits similar wetting behavior, but the blood spreading area on the gauze surface is smaller than that on cotton gauze within a same time period (Fig. 3b). HTMS-g-gauze shows a high hydrophobicity with a WCA of 132.6°. Water and blood droplet neither spreads radially nor diffuses downwards, but stands on its surface. This is due to this gauze is covered by 15-carbon hydrophobic alkyl chains, whose hydrophobicity prevents blood from diffusing into interior. The instant static WCA of USO-g-gauze is ca. 68.2°. Water and blood exhibits a unique and interesting wetting behavior on USO-g-gauze, namely the droplet gradually and vertically diffuses into gauze in 60 s, while the wetted area on surface is almost identical to the size of the droplet. This is totally different from the easy all-directional liquid pervasion on cotton gauze and ABO-g-gauze.

**Water vapor permeation rate and water absorption/movement in gauze**. Hemostatic gauze with proper water vapor transmission and liquid absorption can prevent dehydration and excessive accumulation of exudates. Therefore, they can control water/blood loss and create an ideal moist environment for wound healing[22]. The water vapor permeation rate of cotton gauze, ABO-g-gauze, HTMS-g-gauze, and USO-g-gauze is ca. 1028, 1023, 1021, and 1015 g m$^{-2}$ day$^{-1}$ at 37 °C, respectively (Fig. 3c), suggesting that water vapor permeability of the three surface modified gauzes is as good as that of standard cotton gauze, due to their breathable knitted fabric structure.

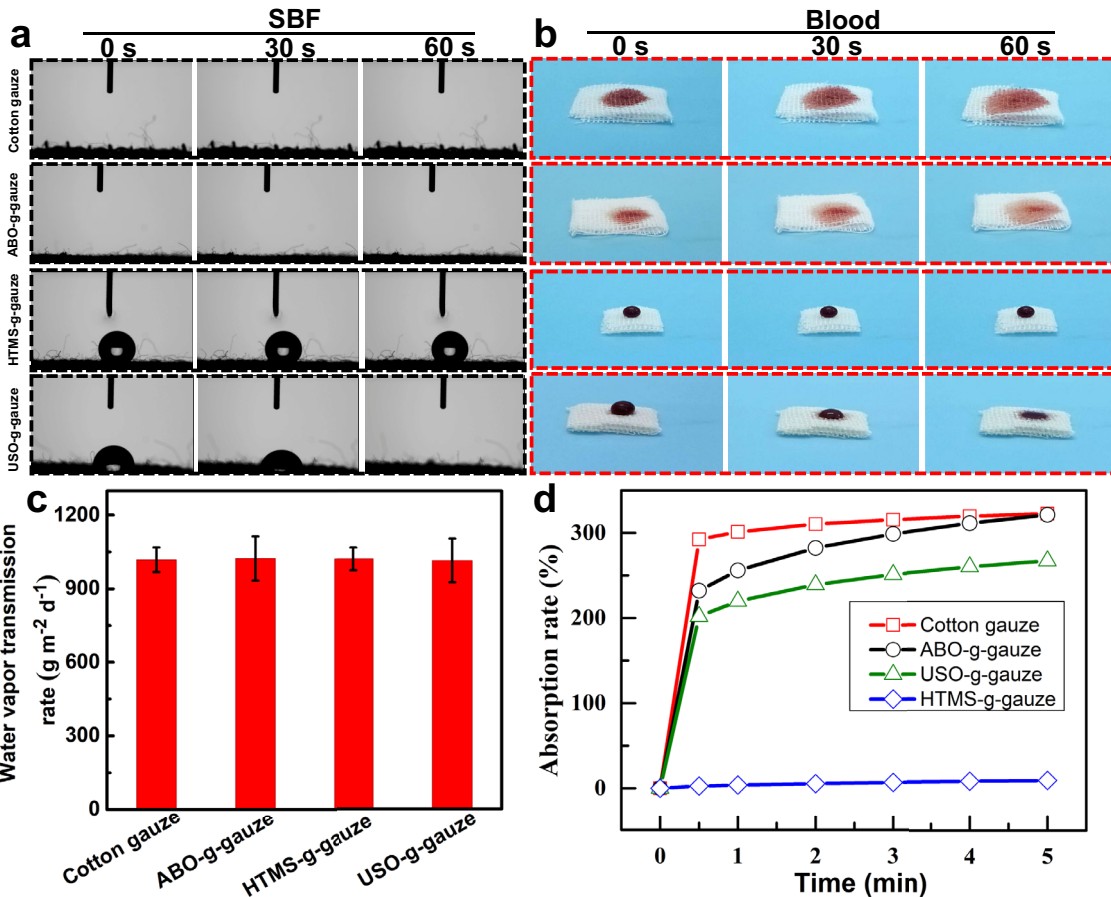

**Fig. 3 Interaction between water and gauzes. a** Water contact angle of gauzes. **b** blood diffusion and spreading in gauzes. **c** Water vapor permeation rate. **d** water absorption rate of gauzes. Data in **c** are shown as mean ± standard deviation (SD), *n* = 3, error bars represent SD. Source data for **c** are provided as a Source Data file.

Cotton gauze and ABO-g-gauze are rapidly wetted by water and sink to bottom when they are put in water. The hydrophobic HTMS-g-gauze always floats on water surface. However, USO-g-gauze shows a very different behavior, i.e., it initially floats on water surface, but sinks after 30 min (Supplementary Movie 1). USO-g-gauze has a thin USO layer on its surface, weakening its hygroscopicity. This lowers its water absorption rate, which is proven by its water absorption dynamic (Fig. 3d). Therefore, it takes a relatively long period for fully swollen to sink. The water absorption rate of cotton gauze, ABO-g-gauze, HTMS-g-gauze, and USO-g-gauze is approximately 323%, 321%, 9.0%, and 267% at 5 min (Fig. 3d), respectively. Compared to cotton gauze, the water absorption rate of USO-g-gauze lose by 17.3%, attributing to the hydrophobic effect from the very thin USO layer on surface. The moisture management test (MMT) indicates that the concomitant hydrophobic/hydrophilic structure of USO-g-gauze imparts it with not only proper wetting time and spreading rate, but its ability of water diffusion from one side to the other side (Supplementary Table 2). These properties would be very helpful for controlling blood movement in gauze and at the gauze/tissue contact surface when it is practically applied as a topical hemostat, as will be shown in the following sections.

**Erythrocyte and platelet adhesion on gauze.** Aggregated erythrocytes and platelets are essential components of blood clot for bleeding control[23]. SEM was used to explore the interaction of erythrocyte/platelet with gauzes. As shown in Fig. 4, erythrocytes adhere on gauze fiber surface to form aggregates. The unique

double concave disk structure of erythrocyte is well maintained, indicating that gauzes don't affect the normal physiological state of erythrocytes. However, the amounts of erythrocyte adhering on gauze fiber surface depend on its wettability. Due to the hydrophobicity of HTMS-g-gauze, fewer erythrocytes are observed on it than on the other three gauzes (Fig. 4c). Cotton gauze accumulates erythrocytes simply by blood fluid absorption, leading to adherence of erythrocytes to its surface (Fig. 4a). However, many erythrocytes aggregate on the fiber surface of ABO-g-gauze and USO-g-gauze (Fig. 4b, d). In addition, the formation of larger platelet aggregations can be observed on USO-g-gauze than on cotton gauze (Fig. 4e, f). It is clearly seen that some platelets with pseudopods are present on USO-g-gauze surface, indicating that platelets have been successfully activated upon contact with gauze fiber[24]. The above phenomena suggest that USO-g-gauze can catch erythrocytes and platelets in plasma to help form blood clots.

**Bleeding control efficiency in rat femoral artery injury model.** The hemostasis efficacy of gauzes on in vitro trauma was evaluated by using the rat femoral artery injury model. The hemostatic performance of the five gauzes is significantly different from each other (Fig. 5a and Supplementary Movie 2). Due to high hydrophilicity, blood quickly permeates into the outmost layer of the stacked four-layer cotton gauze within 1 s. The blood pervasive area on the gauze swiftly expands with increasing time, and the gauze surface is almost fully blood-stained in 60 s. Blood is once again welling out from the wound when the gauze is removed after bleeding stops in 5 min. On the unfolded gauze, the

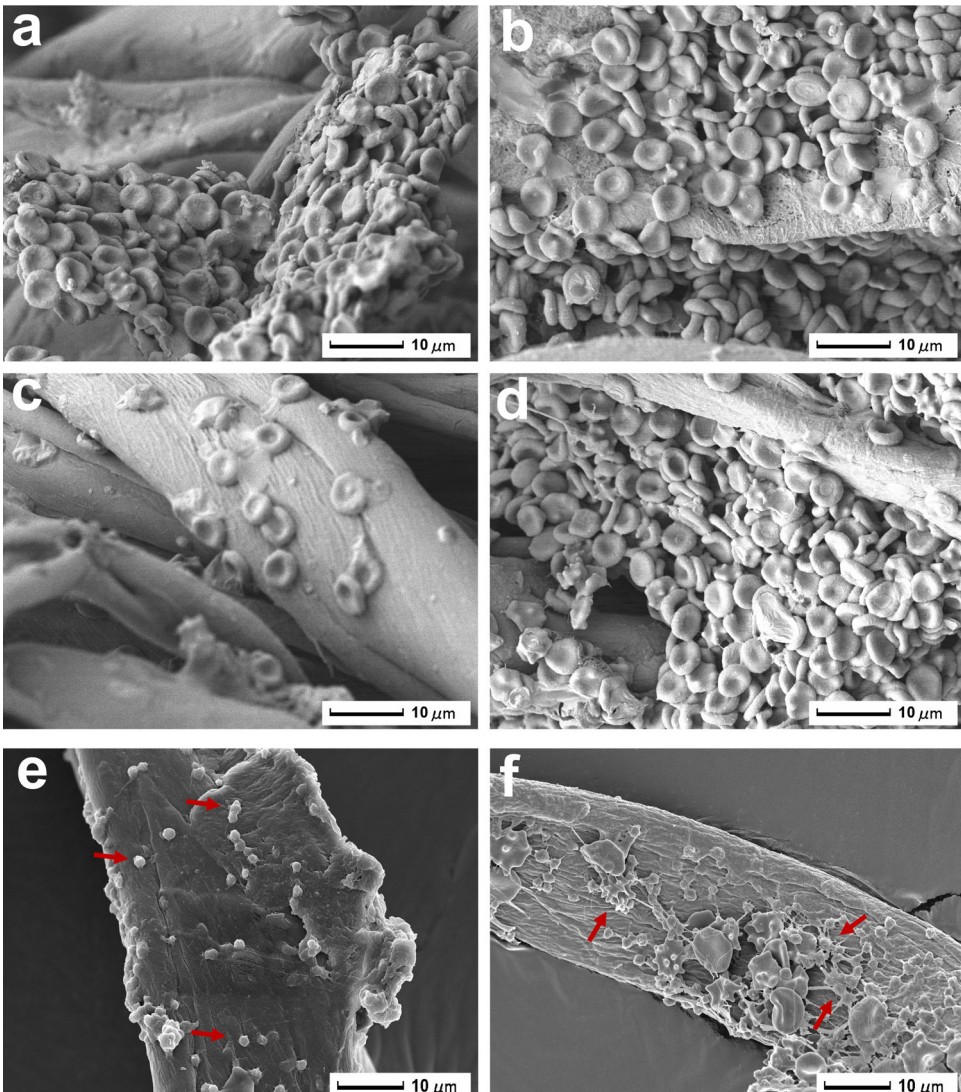

**Fig. 4 Adherence of erythrocyte and platelet on gauze.** Erythrocytes on: **a** cotton gauze, **b** ABO-g-gauze, **c** HTMS-g-gauze, and **d** USO-g-gauze. Platelets (pointed by red arrows) on: **e** cotton gauze fiber, and **f** USO-g-gauze fiber. Three spots were observed independently with similar results.

four gauze layers are completely wetted by blood (Fig. 5-unfolded gauze). These phenomena imply that it absorbs a large volume of blood before bleeding is controlled, and the thrombus is easily destroyed when gauze is removed from the wound (Fig. 5a). The hemostatic performance of ABO-g-gauze is similar to that of cotton gauze, i.e., blood also quickly permeates and wets the whole four-layer ABO-g-gauze. However, the area wetted by blood on it is slightly smaller than that on cotton gauze in 60 s, and the red color of the bloody ABO-g-gauze is lighter than that of cotton gauze (Fig. 5a). On the unfolded ABO-g-gauze, the piles of the four gauze layers are fully permeated by blood, which is like the case of cotton gauze (Fig. 5—unfolded gauze). After ABO-g-gauze is removed post-hemostasis, re-bleeding occurs. However, the blood-stained area around wound is obviously less than that in the cotton gauze group (Fig. 5a), suggesting that the hemostatic performance of ABO-g-gauze is improved to some extent.

When the hydrophobic HTMS-g-gauze comes into contact with trauma, blood does not diffuse vertically into the top gauze layer, but it steadily seeps out of the seam of gauze/tissue surface. On the unfolded gauze, only a small area of the first gauze layer in contact with the wound is stained by blood in 60 s (Fig. 5—unfolded gauze). The gauze is taken away in 10 min when

bleeding ceases, it is seen that blood stains the vicinity of the wound and even wets the gauze underneath the thigh, and severe re-bleeding occurs (Fig. 5a), indicating its hemostatic ability is far worse than cotton gauze. For comparison, when HTMS-g-gauze is compressed onto the bleeding wound, blood spills and still seeps out even after 2-min-compressing (Supplementary Movie 3). This is due to the hydrophobic HTMS-g-gauze has a poor moisture management ability, which is not helpful for hemostat. When applied to a bleeding trauma, the hydrophobic HTMS-g-gauze inhibits blood wetting, absorption, wicking and diffusion into the upper gauze layer (Fig. 3 and Supplementary Table 2), resulting in poor hemostatic performance. As for QCG gauze, an acclaimed gauze for effective controlling of severe bleeding, the outflowing blood diffuses into the four-layer gauze in 1 s, but the blood-stained area on the outmost layer gauze is smaller than that on cotton gauze and ABO-g-gauze. The unfolded gauze, which is removed from the post-hemostatic wound, has blood-stains on all four gauze layers (Fig. 5—unfolded gauze). The wound re-bleeds rather than maintains hemostatic state (Supplementary Movie 2). Very surprisingly, when USO-g-gauze is in contact with the bleeding wound, blood neither diffuses through to the top gauze layer nor seeps out of the seam of gauze/wound surface no matter

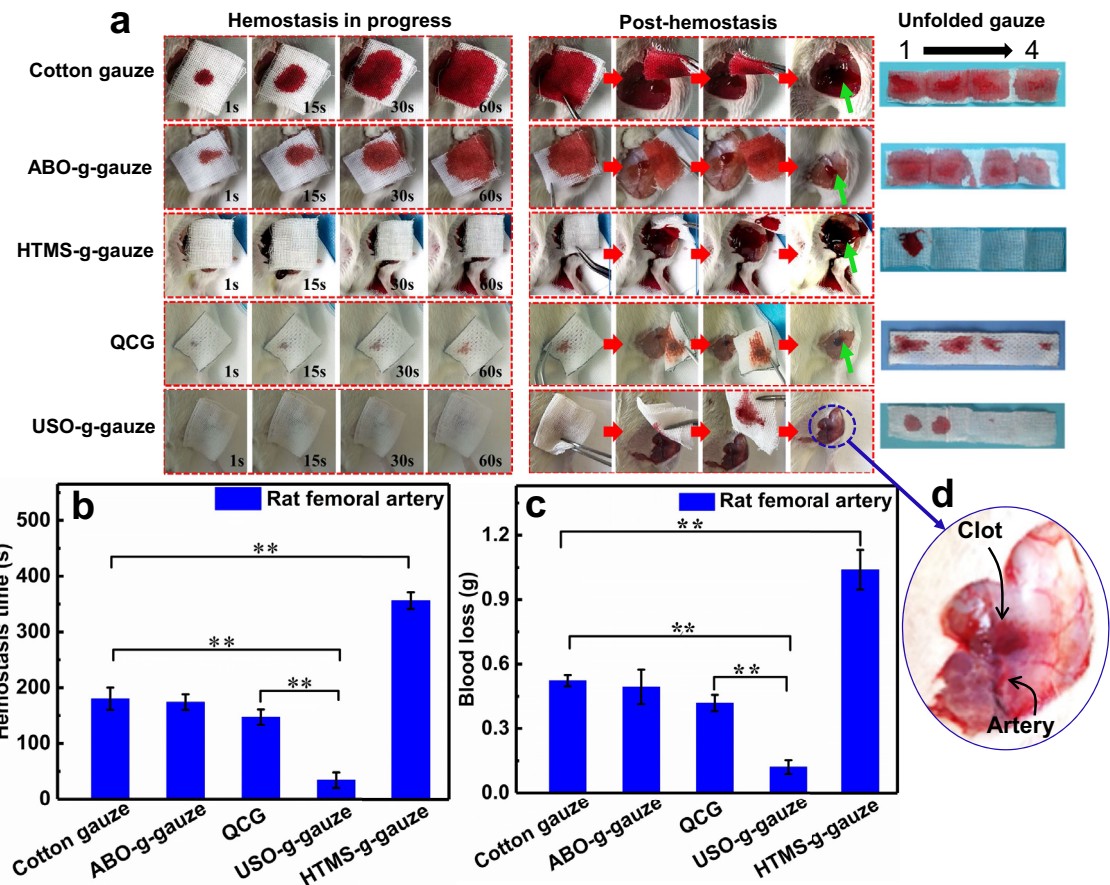

**Fig. 5 Hemostatic performance of gauzes on the rat femoral artery injury. a** From left to right: gauze was put on the bleeding rat femoral artery injury, gauze was removed from the wound after hemostatic state was reached; The stacked four gauze layers were unfolded. "1" was the layer directly contacted with the wound, "4" was the outmost layer. **b** Hemostatic time, and **c** total blood loss in the rat femoral artery model. **d** Enlarged photo of rat femoral artery injury after hemostasis. This photo was taken right after the gauze was removed. Data in **b** and **c** are shown as mean ± SD, $n = 6$, error bars represent SD. Source data for **b** and **c** are provided as a Source Data file. \*\*$P < 0.01$, one-way analysis of variance (ANOVA).

the gauze is applied with (Supplementary Movie 3) or without compressing (Supplementary Movie 2). Compared with the other five types of gauze, there is only one small blood-stain on the first two gauze layers of four-layers in total (Fig. 5—unfolded gauze). In the post-hemostasis, it is amazingly found that the wound and its vicinity is dry and clean without obvious blood-stain, and no re-bleeding occurs. These exceptional phenomena strongly indicate that the USO-g-gauze has excellent hemostatic efficiency on the rat femoral artery injury. In an extended similar work (Section 5 of Supplementary Information), we found that chitosan nonwoven whose surface was slightly grafted with USO also showed significantly enhanced hemostatic performance similar to USO-g-gauze, namely controlled blood spreading and diffusion in gauze, no re-bleeding upon gauze removal, and small blood loss, compared to the performance of pristine chitosan nonwoven gauze (Supplementary Fig. 6 and Movie 4).

On the rat femoral artery injury model, HTMS-g-gauze has the longest hemostasis time of ca. 356 s; The hemostasis time of ABO-g-gauze is ca. 173 s, down from ca. 180 s of cotton gauze; that of USO-g-gauze sharply fells to ca. 34 s from 147 s of QCG gauze (Fig. 5b). Compared with QCG gauze, the hemostatic time of USO-g-gauze drops by 77%. Accordingly, the blood loss follows a decreasing order of HTMS-g-gauze (1.03 g) > cotton gauze (0.52 g) ≈ ABO-g-gauze (0.49 g) > QCG gauze (0.42 g) >> USO-g-gauze (0.13 g) (Fig. 5c). Thus, the blood loss of USO-g-gauze is impressively reduced by about 71%, as compared to QCG gauze.

**Hemostasis in rat liver injury model.** The gauze's hemostatic performance in a non-compressible wound is evaluated by using the rat liver laceration model. Their behaviors of blood diffusion, flow at the gauze/tissue contact surface, and re-bleeding are similar to the hemostasis on the rat femoral artery injury model (Fig. 6a—unfolded gauze, Supplementary Movie 5). Obviously, the hemostatic efficacy of USO-g-gauze is significantly better than the other four gauzes. When USO-g-gauze is removed from the wound in post-hemostasis, no fresh blood wells out from the liver wound. The hemostatic time of cotton gauze, ABO-g-gauze, HTMS-g-gauze, QCG gauze, and USO-g-gauze is ca. 172 s, 153 s, 344 s, 96 s, and 32 s, respectively (Fig. 6b). In addition, the blood loss of cotton gauze, ABO-g-gauze, HTMS-g-gauze, QCG, and USO-g-gauze is ca. 0.39 g, 0.37 g, 0.98 g, 0.13 g, and 0.03 g, respectively (Fig. 6c). Compared with QCG gauze, the hemostatic time and blood loss of USO-g-gauze is reduced by 67% and 77%, respectively. On this injury model, its hemostatic efficacy is also much superior to Surgicel®. Although blood hardly diffuses into the outmost layer, much blood seep out of the seam of Surgicel/liver and stain the cotton gauze under the liver, resulting in hemostatic time and blood loss of ca. 198 s and 0.47 g, respectively (Supplementary Fig. 7c and Movie 5). Therefore, the high hemostatic ability of USO-g-gauze on the non-compressible liver injury is as impressive as on the compressible artery injury model.

In both rat femoral artery injury and liver laceration models, the survival rate within 120 min varies from one gauze to another. It is 100% for rats treated with USO-g-gauze, compared to 20%

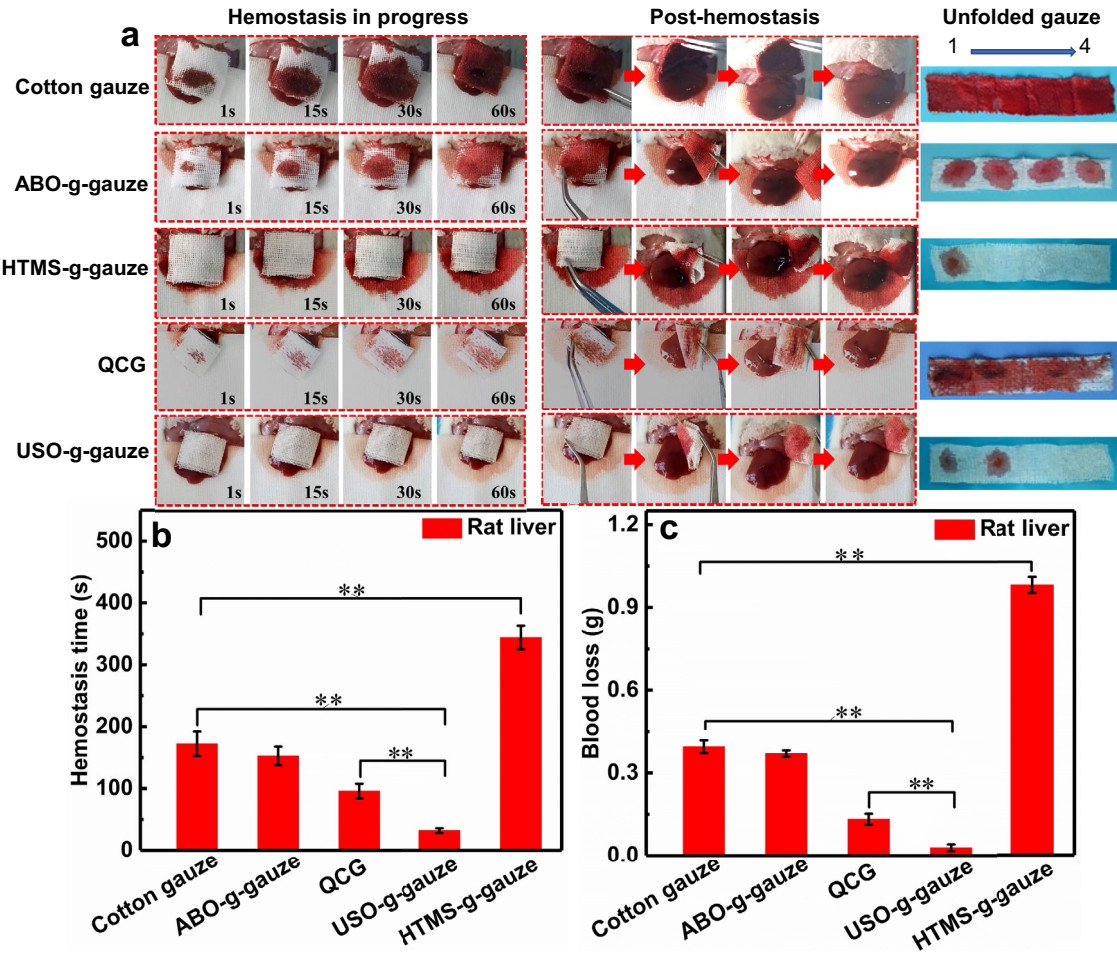

**Fig. 6 Hemostatic performance of gauzes on the rat liver injury. a** From left to right: gauze was put on the bleeding rat liver injury, then it was removed from the wound after hemostatic state was reached; The stacked four gauze layers were unfolded. "1" was the layer directly contacted with the wound, "4" was the outmost layer. **b** Hemostatic time, and **c** total blood loss of gauzes in the liver injury models. Data in **b** and **c** are shown as mean ± SD, $n = 6$, error bars represent SD. Source data for **b** and **c** are provided as a Source Data file. **$*P < 0.01$, one-way analysis of variance (ANOVA).

for QCG, and to no survival for cotton, ABO-g-gauze and HTMS-g-gauze (Supplementary Fig. 8a, b). The survival rate is in accordance with the hemostatic efficacy, i.e., blood loss and hemostasis time of the gauzes as demonstrated in Figs. 5 and 6.

**Hemostatic performance in pig femoral artery injury model.** In order to further evaluate the hemostatic performance of gauzes on massive bleeding wounds, the pig femoral artery injury model is used. The cotton gauze is wrapped around the wound for 3 min, then opened up to check the wound. As shown in Fig. 7a and Supplementary Movie 6, the outmost cotton gauze layer is immediately wetted by blood upon contact with the trauma. The blood diffusion area steadily increases as time goes by. Fresh blood continues flowing out from the wound when the cotton gauze is removed after 3 min (Fig. 7c). In contrast, USO-g-gauze shows far better hemostatic performance (Supplementary Movie 7). Only a small blood-stain is observed on the outmost gauze layer in 60 s (Fig. 7b). Three min later, re-bleeding does not happen upon the gauze is removed from the wound (Fig. 7d). Because this is a severe bleeding wound, a relatively big area is blood-wetted in the first layer, but the 4th layer is stained by a small blood domain only. This suggests blood diffusion in the vertical and radial directions is largely restricted in USO-g-gauze. It should be pointed out that the accurate hemostatic time of every gauze is not measured on this injury model, because it is

hard to judge when the wound stops bleeding due to the wound is wrapped by gauze and re-bleeding often occurs upon uncovering wound. Therefore, at the time interval of 3 min, the blood mass absorbed by gauze is measured to roughly reflect the blood loss. This is ca. 0.80 g for USO-g-gauze, while that of cotton gauze, ABO-g-gauze, QCG, and HTMS-g-gauze is 5.11 g, 4.16 g, 3.94 g, and 8.22 g, respectively (Fig. 7e). Therefore, the blood loss from the pig femoral artery wound treated with USO-g-gauze is only 15.6% and 20.4% of that with cotton gauze and QCG, respectively. In fact, the hemostasis time of USO-g-gauze on this wound model is less than 3 min, as suggested by the fact that no occurrence of re-bleeding upon uncovering the wound (Fig. 7d).

Hemostasis on the pig skin laceration model is also examined since its structure is very similar to human skin. A regular cut with a length of 2 cm and a depth of 1 cm was made with a scalpel, then a four-layer gauze was applied onto the wound. The dynamic hemostatic process of the four gauzes demonstrates that blood diffusion and absorption, and blood flowing underneath gauze, are very similar to the hemostasis on the rat femoral artery injury and liver laceration models. Blood even rarely stains the hydrophobic HTMS-g-gauze, but oozes out from the seam of gauze/skin surface (Supplementary Fig. 9). The blood loss of the cuts treated with cotton gauze, ABO-g-gauze, QCG, HTMS-g-gauze, and USO-g-gauze is ca. 0.54 g, 0.32 g, 0.22 g, 0.71 g, and 0.03 g (Fig. 7f), respectively. Compared with cotton gauze

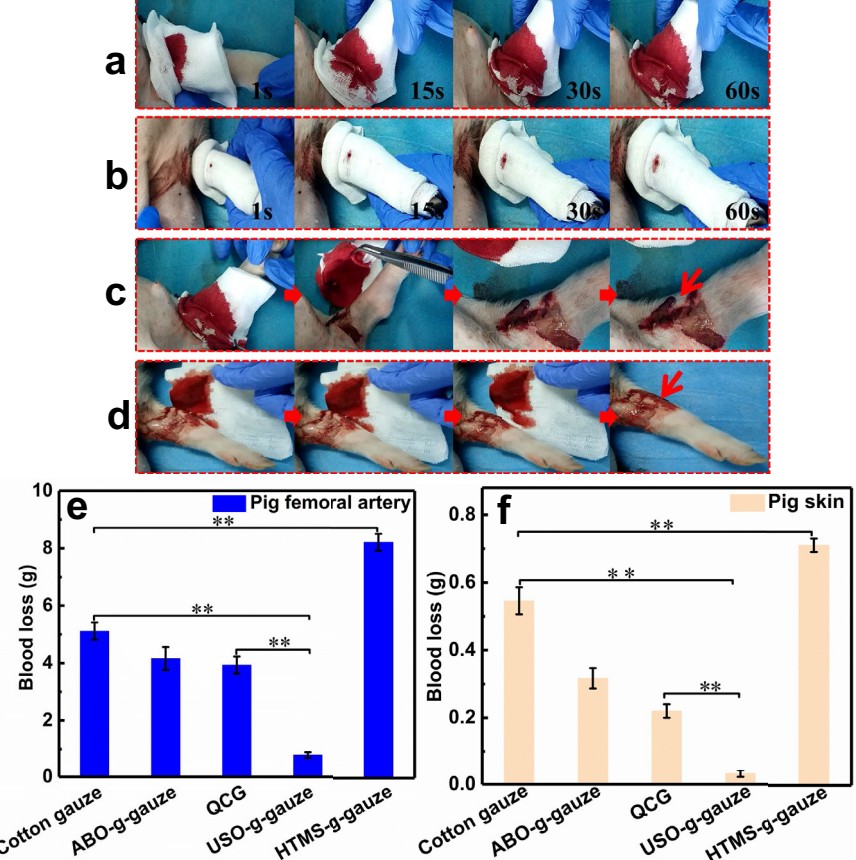

**Fig. 7 Hemostasis in the pig femoral artery injury models.** Hemostasis process of **a** cotton gauze, and **b** USO-g-gauze. The status of wound after **c** cotton gauze, and **d** USO-g-gauze were removed after it was treated for 3 min. Blood loss in the pig **e** femoral artery, and **f** skin injury models. Data in **e** and **f** are shown as mean ± SD, n = 3, error bars represent SD. Source data for **e** and **f** are provided as a Source Data file. **P < 0.01, one-way analysis of variance (ANOVA).

and QCG, the blood loss for USO-g-gauze reduces by ca. 94% and 85.5%, respectively. The results of the pig femoral artery injury and skin laceration models further justify that USO-g-gauze has excellent hemostatic efficacy for severe bleeding wounds.

**Hemostatic mechanism of the highly efficient USO-g-gauze.** In each injury model, the bleeding wounds treated with ABO-g-gauze is cleaner, and the blood diffusion area on gauze and blood loss are smaller than those treated with standard cotton gauze. ABO-g-gauze not only concentrates blood components due to its quick water absorption ability (Fig. 3d), but also catches blood cells by its tissue adhesive catechol groups (Fig. 4b), so the hemostatic efficiency of ABO-g-gauze is slightly improved. The hemostatic performance of HTMS-g-gauze is significantly inferior to that of cotton gauze, because of strong repellence of blood fluid by the highly hydrophobic HTMS alkyl chain (Figs. 3 and 4c). However, the USO-g-gauze containing a hydrophobic long alkyl chain with a tissue adhesive catechol end group exhibits impressively excellent hemostatic efficacy.

Why has USO-g-gauze the most excellent hemostatic capability among those gauzes? To better understand the nature behind this feature of USO-g-gauze, we initially carried out the density functional theory (DFT) calculations to investigate the adsorption interaction of sixteen different kinds of amino acids (they are essential components of tissue keratin protein[25]) with USO-g-gauze, as illustrated in Supplementary Fig. 10. More details have been provided in Section 10 of Supplementary Information. The adsorption energies ($\Delta E_{ads}$) of these amino acids to USO-g-gauze

are calculated by considering the main non-covalent interaction modes including π–π stacking and hydrogen bond interactions.

Our computed results reveal that the amino acids containing π-conjugated benzene ring, such as phenylalanine (F) and tyrosine (Y), can be effectively adsorbed on the catechol of USO-g-gauze through synergistic actions of π–π stacking and hydrogen bonding (Fig. 8c and Supplementary Fig. 10), where the computed $\Delta E_{ads}$ values are as large as 0.621 and 0.729 eV, respectively. Comparatively, all the remaining fourteen amino acids without π-conjugated ring can effectively interact with the catechol of USO-g-gauze by adopting double hydrogen bonds, as presented in Fig. 8c and Supplementary Fig. 10. All these hydrogen bonding distances are in the range of 1.69 ~ 1.80 Å (Supplementary Table 3), and the calculated $\Delta E_{ads}$ values are as big as 0.570–0.639 eV (Supplementary Fig. 10), indicating strong interaction force between them. Further analyses demonstrate that the unique structure of catechol group plays a crucial role in forming double hydrogen bonds or the synergistic action of π–π stacking and hydrogen bonding (Supplementary Figs. 11 and 12). These non-covalent interactions contribute to USO-g-gauze's strong tissue adhesiveness.

Besides the above molecular level analyses of adhesion interaction, the adhesion force (or peeling force) of those gauzes on fresh rat femoral tissue is measured and shows obvious variation from one to another. As expected, the hydrophobic HTMS-g-gauze has the lowest peeling force of 24 mN, while USO-g-gauze shows the largest force of 90 mN, which is ca. two times as much as that of cotton gauze (Supplementary Fig. 13). From Supplementary Movie 3, the adhesion of ABO-g-gauze and

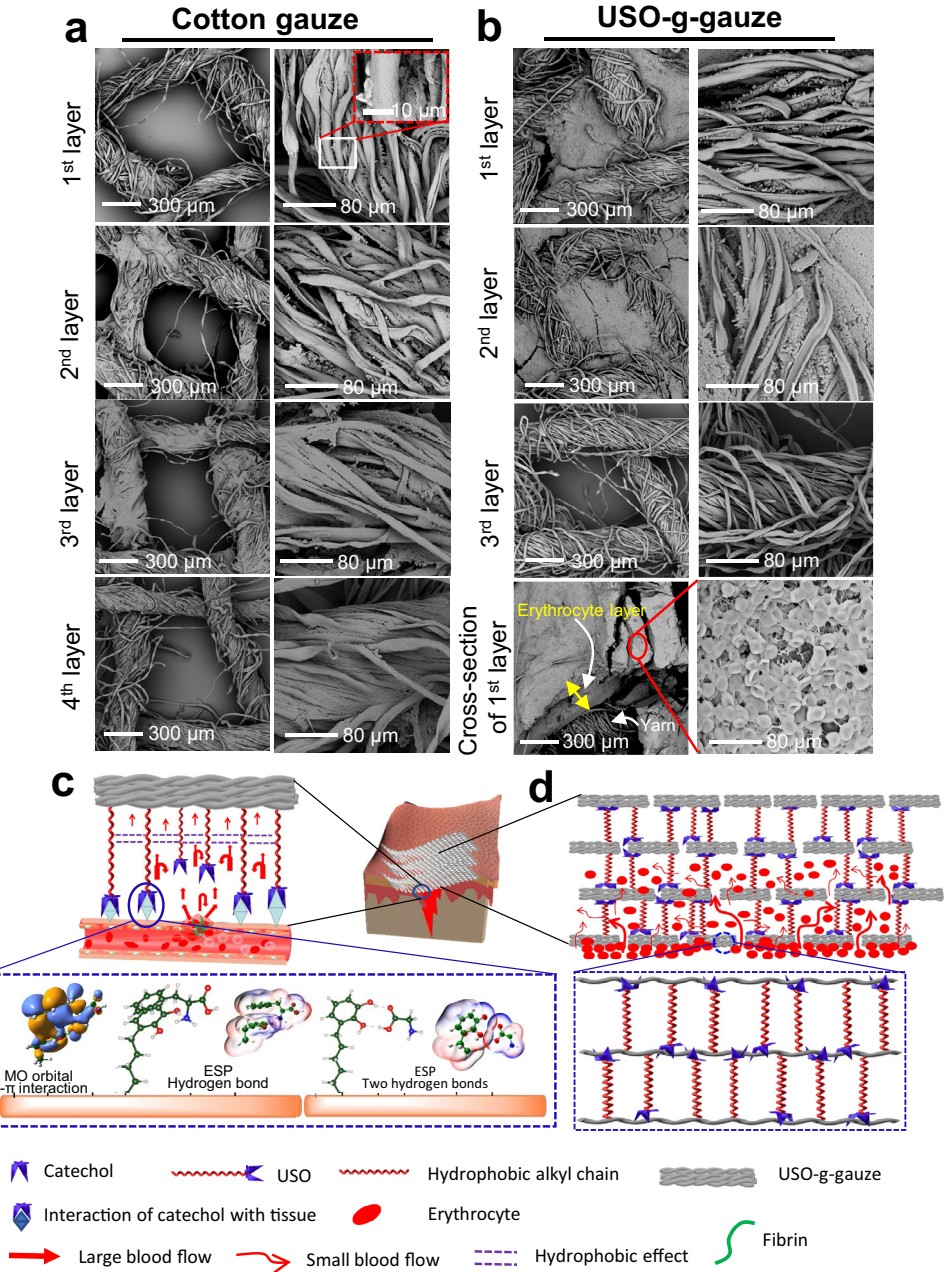

**Fig. 8 Aggregation of erythrocytes on cotton gauze and USO-g-gauze after hemostasis in the rat femoral artery injury model; and hemostatic mechanism diagram of USO-g-gauze. a** On cotton gauze patch composed of four stacked gauze layers, erythrocytes sparsely distribute on cotton yarns of all layers. The macropores among yarns are vacant. **b** On USO-g-gauze patch composed of four stacked gauze layers, erythrocytes fill the macro-pores among yarns in the first two layers, but are absent in the pores and on the yarns of the 3rd layer. Cross-section of the 1st layer shows a thick erythrocyte layer. **c** Through adhesive bonds like π–π stacking interaction and hydrogen-bond between USO's catechol group and wound tissue's amino acid units, dam-like barriers form surrounding the wound. They retard blood seeping out of the tissue surface. The repelling pressure from the hydrophobic effect among long alkyl chains slows down blood wicking movement. **d** Dam-like barriers forming by USO between gauze layers and between fibers, retard blood diffusion in the vertical and horizontal directions, largely confine blood movement in the pores between warp and weft yarns, resulting in large accumulation of erythrocytes. Images in **a** and **b**, five spots were observed independently with similar results.

USO-g-chitosan to wound tissue is obviously perceivable when they were peeled off from the wounds, while it is less noticeable in the cases of cotton gauze etc (Supplementary Movie 2). This vividly confirms the existence of adhesive interaction between catechol and tissue, but such non-covalent adhesion can be broken with mild peeling forces. In the case of the double −OH groups of catechol were modified such as with chelation with $Fe^{3+}$ or oxidation into quinone, the tissue adhesion force sharply

decreases to values close to that of HTMS-g-gauze (Supplementary Fig. 13). This further confirms that catechol group plays an essential role in the wet tissue adhesion[26–28].

In order to demonstrate the importance of tissue adhesion of catechol to high hemostatic efficiency, the catechol group on USO-g-gauze is transformed to lower its tissue adhesiveness. Hence, USOFe-g-gauze and USOQu-g-gauze were fabricated. On USOFe-g-gauze, catechol groups readily react with $Fe^{3+}$ to form a

complex, while on USOQu-g-gauze the catechol groups are oxidized into quinone (Supplementary Fig. 14a). Both catechol-$Fe^{3+}$ and quinone groups have no or weak adhesion to wet tissue[29]. The results show that the hemostatic efficacy of USOFe-g-gauze and USOQu-g-gauze on rat injury models decreases substantially (Supplementary Figs. 15 and 16).

Therefore, USO, a catechol compound with a side alkyl chain having 15 carbons is a good candidate compound for surface modification of fabric gauze (such as cotton gauze and chitosan nonwoven) to prepare highly efficient hemostatic gauzes. Certainly, other properties of USO-g-gauze such as its moisture management ability are also essential to the high hemostatic efficiency. The blood fluid movement in gauze and around the gauze/tissue contact surface governed by the unique wetting property and tissue/cell adhesiveness facilitates aggregation of massive erythrocytes, as shown in Fig. 8b. There are so many congested erythrocytes that they even fill the quadrilateral macropores among warp and weft yarns of the first two gauze layers (the layer directly contacts with tissue is the first layer), but none in the 3rd layer, which are consistent with the observation shown in Figs. 5 and 6—unflolded gauze. The thickness of the erythrocyte layer accumulated on the 1st layer reaches as high as 220 μm. As well-known, erythrocytes are the key component of the primary blood clot. Thus, more erythrocytes are aggregated, bigger clot is formed, shorter bleeding time and less blood loss are attained. Therefore, the thick erythrocyte layers in the first two USO-g-gauze layers serve as clots for effectively controlling bleeding. However, the erythrocyte accumulation ability of cotton gauze is poor as suggested by its sparse distribution on cotton gauze yarns with none in the quadrilateral macropores of the whole four gauze layers (Fig. 8a). This is because erythrocytes move along with the fast blood wicking to everywhere in the cotton gauze patch, rather than group together to form a big erythrocyte plug. But such a movement is retarded in USO-g-gauze by the anchoring hydrophobic chain barriers at the contact surface of USO-g-gauze/tissue and among fibers and yarns.

Therefore, the hemostatic mechanism of USO-g-gauze is proposed in Fig. 8c, d. When USO-g-gauze is applied onto a bleeding wound, catechol groups of USO quickly anchor to skin tissue through non-covalent bonds such as hydrogen bond and π–π interaction, to form dam-like barriers around wound (Fig. 8c). This can hinder and eventually prohibit blood from seepage at the gauze/tissue contact surface. The repelling pressure from the massive hydrophobic interaction among the long alky chains retards blood diffusion into the upper gauze layers, so only the first two layers are blood-wetted as shown in the rat and liver injury models (Figs. 5 and 6—unfolded gauze). Even in the pig injury models, blood has a difficulty in diffusing radially, and only little volume of blood reaches the outmost layer to result in a small blood stain (Fig. 7b). This should contribute to the massive dam-like barriers formed among yarns and fibers due to the interaction between USO catechol and cotton cellulose, hence big blood stream moves in the pores between warp and weft yarns, with small blood streams in other pathways (Fig. 8d). Finally, the blood wicking capability of the moderately hydrophilic USO-g-gauze fibers facilitate concentration of erythrocyte, platelet and protein to quicky form big primary blood clots (Fig. 8d). Thus, the synergistic effects of tissue adhesion, hydrophobic interaction, and hydrophilic fiber structure make USO-g-gauze an excellent hemostatic gauze. The other impressive feature of USO-g-gauze is that no re-bleeding occurs upon removing it from wound after hemostasis, while re-bleeding is often experienced when cotton gauze is used. In the case of cotton gauze, with sparse aggregation of erythrocyte on yarns, it is an essential part of the blood plug (Fig. 8a). The plug would be easily broken since erythrocytes are removed along with peeling-off cotton gauze, leading to secondary bleeding. From Fig. 8b, it's seen

that the erythrocytes accumulated at the injury site is so enormous and thick that removal of USO-g-gauze would take away part of erythrocytes, but some erythrocytes (Fig. 5d) remain on site to avoid re-bleeding. The mechanism for no re-bleeding of USO-g-gauze is different from that of Bandage®, which is a well-known no re-bleeding hemostatic fabric strip for small bleeding wounds. Bandage's anti-adhesion relies on the hydrophobic membrane covering on the water-absorbent fabric layer (Supplementary Fig. 17). The poor adhesion of hydrophobic membrane to fibrin prevents bandage from being a part of blood clot. The no re-bleeding is actually very important when wounded person are relocated or injured tissue/organs are moved accidently, so fresh blood loss can be substantially avoided. Since the hemostasis of USO-g-gauze is a physical blocking effect rather than change of the body's normal physiologic clotting mechanisms, it would also show hemostatic efficacy for patients with coagulopathy.

**Biocompatibility of USO-g-gauze**. The in vitro cytocompatibility assay shows USO-g-gauze is non-cytotoxic, same as cotton gauze (Supplementary Figs. 18 and 19). Figure 9 shows histological changes of the subcutaneous muscle tissue treated with cotton gauze and USO-g-gauze at specific time points (3, 7, 14, and 21 days) using H&E staining and toluidine blue staining. In the cotton gauze-treated tissue, plenty of neutrophils arise around the gauze on the third day after implantation (Fig. 9a) and the corresponding number are counted to be about $50 \pm 5$ (Fig. 9c). However, very few neutrophils are observed 7 days later, which reduces to about $2 \pm 1$ after 21 days. Meanwhile, USO-g-gauze has a similar inflammatory response. The density of neutrophils increases in the USO-g-gauze treated tissues since implantation, but rapidly decreases to $4 \pm 1$ after 7 days. In addition to neutrophils, mast cell is another critical effector of inflammation. As shown in the toluidine blue staining (Fig. 9b, d), several mast cells are observed in the tissue section contacting with the cotton gauze and the USO-g-gauze on the seventh days after implantation, but sharply reduce 14 days later within the cotton and the USO-g-gauze treated tissue. The subcutaneous implantation examinations reveal that the surface modification with USO doesn't compromise the biocompatibility of cotton gauze, and causes no significant inflammatory responses. It should be mentioned that the free small catechol compound USO may cause contact dermatitis according to the US CDC and EPA, but the USO immobilized on cotton gauze through surface grafting has shown no dermatitis to our derma so far. In fact, polymerized USO has low cytotoxicity for dentistry application[30] and has a long history as a safe coating for wooden bowl in oriental countries.

In summary, a highly efficient hemostatic cotton gauze was developed through simply grafting a catechol compound USO (with a long hydrophobic alkenyl side chain) onto cotton gauze surface. The special pendent groups offer wet tissue adhesiveness and hydrophobicity to the gauze. On basis of findings of the DFT calculation and measurements, a new physical hemostatic mechanism is proposed for this gauze, namely many dam-like barriers are formed at the contact surface of gauze/wound tissue and among adjacent fibers. These barriers are capable of retarding blood flowing and controlling the movement of bloodstream at the seam of gauze/wound contact surface and in the gauze, resulting in quick aggregation of erythrocyte to form a thick blood clot for stopping bleeding. Thus, this gauze can overcome the intrinsic high blood absorption capacity of conventional cotton gauze, making it to significantly reduce additional blood loss from bleeding traumas, and therefore increasing survival rate and decreasing medical such as transfusional costs. Excellent hemostatic efficacy (including short hemostatic time, low blood loss, and no re-bleeding) of this gauze was observed on the rat femoral artery and liver laceration models,

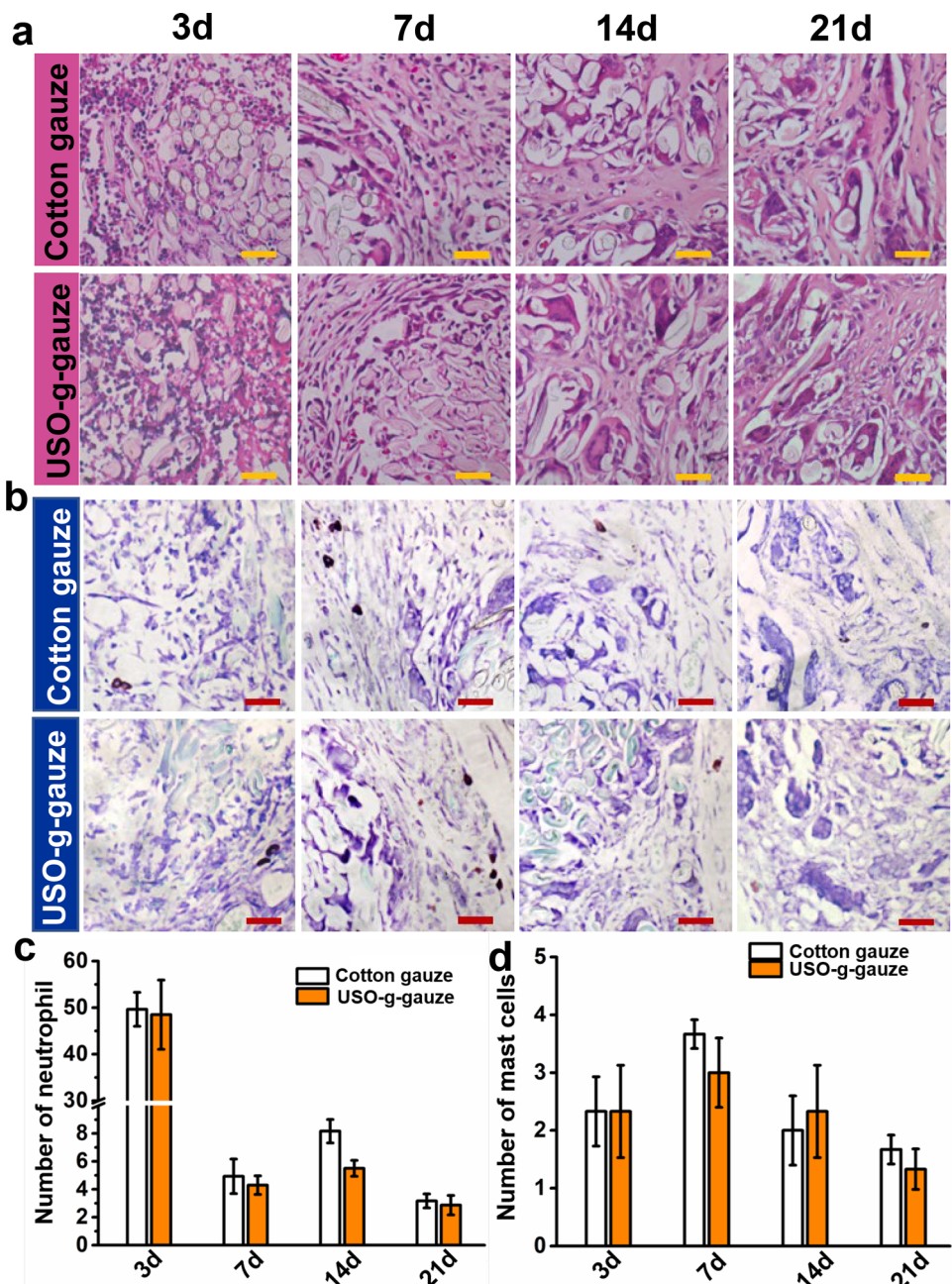

**Fig. 9 Histological changes of the subcutaneous muscle tissue treated with gauzes. a** H&E staining, **b** toluidine blue staining sections of subcutaneous tissue surrounding the gauzes at varied time intervals in the muscle implantation experiments using rats. The corresponding numbers of: **c** neutrophil, and **d** mast cells at different time intervals. Data in **c** and **d** are shown as mean ± SD, $n = 6$, error bars represent SD. Source data for **c** and **d** are provided as a Source Data file. Scale bar: 40 μm.

pig skin laceration and femoral ar1tery massive bleeding wound. Its hemostatic performance is much superior to standard cotton gauze and QCG. The USO-g-gauze displays similar cell and tissue compatibility to cotton gauze. Such an idea and methodology were also successfully applied to make USO-g-chitosan gauze, whose hemostatic efficacy is much better than the chitosan control. We speculate this magic gauze may find very promising emergency-care and clinical applications for controlling traumatic massive bleeding for wounded soldiers in battlefields, civilians in accidents, patients in operation rooms, and patients with coagulopathy.

## Methods

**Materials**. Sterile cotton gauze and chitosan nonwoven fabric were purchased from the commercial market in China. Surgicel®, a knitted fabric gauze of oxidized regenerated cellulose, was from Ethicon Inc. Sodium alginate (SA), acetic acid, glutaraldehyde (GA), paraformaldehyde, and phosphate buffer solution (PBS) were bought from Sinopharm Group Chemical Reagent Co., Ltd. QuikClot Combat Gauze (QCG) was provided by Z-Medica and used as received. The catechol compound USO was extracted from the raw lacquer provided by Xi'an Raw Paint Research Institute by a reported procedure[31]. Briefly, Chinese lacquer sap (1.0 kg) was dissolved in ethanol (2.0 L) under mechanically stirring for 24 h at room temperature, followed by filtration to collect filtrate. From it USO was obtained after ethanol was removed by vacuum-rotary evaporation at 60 °C. Eugenol, trie-thylsilane (TES), tris(pentafluorophenyl)borane (TPFPB), 3-(4,5-dimethyl-2-thia-zolyl)-2,5-diphenyl-2-H-tetrazolium bromide (MTT), and HTMS were purchased from Aladdin Reagent Co., Ltd. All chemicals were used as received without further purification. ABO was prepared according to the procedure described in Supplementary Fig. 1. Male Sprague–Dawley (SD) rats (6–7 weeks old) with weight of 200–250 g were bought from Shanghai Slack Experimental Animal Co., Ltd, China. L929 cells were ordered from ATCC (Catalog. No. C054), and used directly without authentication.

**Preparation of USO grafted cotton gauze (USO-g-gauze).** The preparation scheme and chemical reaction of USO-g-gauze are shown in Fig. 1. Cotton gauze was sequentially washed with distilled water and ethanol, two times in each solvent and 30 min each. Then it was dried under nitrogen stream. To introduce free radicals onto gauze, it was treated by a low temperature plasma in $N_2$ at 400 Pa, 80 W for 3 min (PT-5S, Sanhe Poda Co., Ltd, China). The preparation conditions for USO-g-gauze were optimized (Supplementary Table 1 and Supplementary Fig. 2). In an ideal condition, USO-g-gauze was fabricated by placing the plasma-treated gauze in a mixture of 2.0 wt% USO/ethanol and refluxing at 70 °C for 2 h. Then, after washed 3 times with ethanol, it was dried in a vacuum oven at 80 °C for 2 h to obtain USO-g-gauze. About 0.1 wt% USO was grafted, as determined by the gravimetrical method.

The ABO grafted gauze (ABO-g-gauze) was made by the same way as that for USO-g-gauze, except that ABO was used to replace USO in the reaction mixture (Fig. 1b). HTMS with a long alkyl chain in its chemical structure was grafted onto cotton gauze to make HTMS grafted gauze (HTMS-g-gauze) in a similar way for preparing USO-g-gauze (Fig. 1c).

**Characterization.** The morphology of gauzes was observed by scanning electron microscopy (JEOL-7500LV, Japan). The solid-state $^{13}C$ NMR spectra were characterized by a superconducting fourier transform nuclear magnetic resonance spectrometer (Bruker Avance III 400 WB, AVANCE III, Switzerland). X-ray photoelectron spectra was measured by scanning XPS microprobe instrument (Thermo K-Alpha+, UK). Fourier transform infrared (FTIR) spectra in KBr form were obtained on an FTIR spectrometer (PerkinElmer, 1600) in the range of 4000–400 $cm^{-1}$. The wettability was determined by DSA 100 (Krüss, Germany).

**Moisture management test.** The tests (wetting time, wetted radius, water absorption rate, water spreading speed, and cumulative one-way transport capacity) were performed on a moisture management tester (M290, SDL ATLAS, USA) according to AATCC 195 by measuring the electrical resistance of the top and bottom sides of the gauze[32]. The gauze size was 5 cm × 5 cm. 0.2 mL of standard test solution was dropped onto the gauze in the test. Three replicates were run for each gauze.

**Water vapor transmission rate and water absorption capacity of gauze.** To measure water vapor permeability, a beaker containing 50 mL of distilled water was covered with a gauze. The circumferential border was tightly sealed to prevent any water vapor loss through the boundary. The water vapor transmission rate ($W_{evap}$) was determined by measuring the mass loss of water in the beaker after 24 h at 37 °C. The $W_{evap}$ (g $m^{-2}$ $day^{-1}$) was calculated according to Eq. (1):

$$W_{evap} = (m_b - m_a)/A \qquad (1)$$

where $A$, $m_b$, and $m_a$ are the area of the beaker mouth ($m^2$), the weight of the beaker before and after water evaporation, respectively.

For measurement of water absorption rate, a square gauze with size of 2 cm × 2 cm was immersed in simulated body fluid (SBF), then at certain time interval it was taken out and placed on a filter paper to absorb free water, followed by weighing its mass. The gauze mass before ($m_b$) and after water absorption ($m_a$) was measured, and water absorption rate ($W_{abs}$) was calculated by Eq. (2):

$$W_{abs} = (m_a - m_b)/m_b \times 100\% \qquad (2)$$

**Hemostasis evaluation.** All animal experiments were carried out in accordance with the guidelines for the protection and use of experimental animals in Fujian Normal University. The experiments were approved by the Animal Ethics Committee of Fujian Normal University (Protocol No. IACUC-20180013). The gauze was tailored into rectangle swatch with a size of 12 cm × 2.5 cm, and four swatches were stacked together before used on a bleeding wound. The hemostatic study included four animal injury models: rat femoral artery and liver injury, pig femoral artery and skin laceration. Six rats or three pigs were randomly selected as a group in each animal model and assigned to each sample. QCG is the standard military hemostatic agent recommended by the Tactical Combat Injury Care Committee for use as a control. Anesthesia was injected intraperitoneally with a 10 wt% chloral hydrate solution (0.4 mL/100 g). After complete hemostasis was reached, all animals were observed for 2 h or until death. The survived animals were euthanized with 10% chloral hydrate at the end of experiment.

**Rat femoral artery injury model.** The proceeding of femoral artery injury model was conducted as follows[33]. Rats were randomly selected and anesthetized. Then the fur on the rat thigh was shaved off to expose the femoral artery. Pre-weighed cotton gauze was placed beneath the thigh. Then the artery was transected. After bleeding for 2 s, pre-weighed gauze was gently applied or compressed onto the trauma (compressing for 150 s). Blood diffusion in the gauze was recorded. Bleeding time and blood loss were measured ($n = 6$). We define that hemostatic stage is reached when the blood-stained area on gauze doesn't expand and no blood seeps out of the seam at gauze/wound contact surface.

For observing the micro-morphology of the blood-stained gauze after hemostasis, a patch stacked with four gauze layers was dressed onto the rat femoral

artery injury to reach hemostasis. The gauze was placed in centrifuge tubes and fixed with 2.5% (v/v%) GA/PBS solution at room temperature for 4 h. Then it was rinsed twice with distilled water, followed by sequential dehydration in 0%, 10, 20, 30, 40, 50, 60, 70, 80, 90, and 100% (v/v%) ethanol/PBS solution. Consequently, it was dried at 37 °C. The gauze was sputter-coated with gold before SEM observation.

**Rat liver laceration model.** The liver laceration model was conducted according to a standard procedure[34]. Briefly, rats were randomly selected and anesthetized. Subsequently, the epithelial tissue of the abdomen was cut to expose the liver, and pre-weighted cotton gauze was placed under the liver. A scalpel was used to make a linear incision trauma of about 1 cm in length on the left lobe of the liver. After bleeding for 2 s, pre-weighed gauze was applied onto the trauma. Blood permeation in the gauze, bleeding time, and blood loss were recorded ($n = 6$).

**Pig skin laceration and femoral artery injury models.** Two-month-old male Bama miniature pigs (weight 1.6–1.8 kg) were used to simulate massive bleeding trauma[35]. Anesthesia was injected intraperitoneally with a 10 wt% chloral hydrate solution (0.4 mL/100 g). After the hairs on the back and leg were shaved off, the following two different traumas were created: (1) A linear incision of 2 cm (length) x 1 cm (depth) was made on the back of the pig with a scalpel; (2) A distally extending 4 cm longitudinal incision was made in the right femur region to expose the femoral artery[36]. Blood loss were recorded ($n = 3$).

**Peeling force of gauze on wet rat femoral tissue.** Rats were pre-treated as in the femoral artery injury model, then the peeling test was conducted before the artery was transected. Gauze (5 cm × 2.5 cm) was immediately put on the exposed fresh femoral tissue and kept there for 10 min. The gauze was then peeled off with a Digital Push Pull Gauge (Locosc Ningbo Precision Technology Co., Ltd., China), and the peak force was recorded ($n = 3$).

**Effect of catechol functional groups on hemostasis.** In order to elucidate the effect of catechol structure on hemostatic efficiency of USO-g-gauze, two models were designed: (1) The catechol groups were protected by chelating with $Fe^{3+}$ ions[37]. Typically, USO-g-gauze was soaked in 100 mL of 0.1 M $FeCl_3$ aq. solution for 10 min at 37 °C to allow the occurrence of coordination reaction between catechol groups and $Fe^{3+}$. After washed with distilled water, it was dried in a vacuum oven at 80 °C for 0.5 h. As-obtained gauze was coded as USOFe-g-gauze. (2) The catechol groups were oxidized to quinone. The USO-g-gauze was treated in 100 mL of 20 mM Tris-HCl buffer solution (pH 9.8) for 10 min at 37 °C to convert the phenolic structure to quinone structure[38]. After washed with distilled water, it was dried in a vacuum oven at 80 °C for 0.5 h. As-obtained gauzed was coded as USOQu-g-gauze.

**Accumulation of erythrocytes and platelets on gauze.** The platelet-rich plasma (PRP) was obtained by centrifuging rat whole blood at 1006×$g$ for 20 min at 4 °C[39]. In a volume ratio of 1:10, the whole blood or PRP was added into PBS (pH 7.4) containing a piece of gauze (1 cm × 1 cm), followed by incubating for 90 min at 37 °C. Subsequently, the gauze was rinsed three times with PBS to remove physically adhered blood cells and platelets, and fixed with 2.5% (v/v%) GA/PBS for 2 h. Then it was rinsed twice with distilled water, followed by sequentially dehydrating with 25, 50, 75, 85, 90, and 100% (v/v%) ethanol/PBS solution. Finally, it was air-dried at 37 °C. The adhesion of erythrocyte and platelet on gauze surface was observed by SEM.

**Biocompatibility of gauze.** The proliferation of L929 cells on gauze was measured by using a live/dead assay kit (Beyotime Biotechnology, Shanghai, China)[40]. Briefly, after treated with 75% medical alcohol and 0.1 mg/mL physiological saline, the gauze was transferred into a 24-well plastic culture plate. One milliliter of fibroblast with a density of $2.5 \times 10^5$ cells/mL was gently added on the surface of the gauze surface and cultured at 37 °C in a humidified atmosphere of 5% $CO_2$ for 1, 2, and 3 days. Then, 10 μL of the combined Live/Dead cell-staining solution (2 μM Calcein-AM and 4 μM PI) was added into 400 μL of culture medium and was incubated at 37 °C for 4 h. Finally, images of the live (green fluorescence) and dead (red fluorescence) cells were obtained using an inverted fluorescence microscope.

**In vivo inflammatory assay.** Rats were maintained under a 12 h light/12 h dark schedule with a continuous supply of food and water. After a week of adaptation, the rats were randomly divided into two groups: cotton gauze group and USO-g-gauze. Each group contained 20 rats. The gauze was cut into a square (1 cm × 1 cm) and were sterilized by ultraviolet irradiation for 2 h. The rats were anesthetized by inhalant anesthetics-ether and fixed on the surgical plate. After the shave and the iodine disinfection, a longitudinal incision about 2 cm were made symmetrically on both sides of the spine. Then, the cotton gauze or USO-g-gauze were respectively implanted into the subcutaneous sac of the back. The incision was sutured by the simple intermittent suture method. Finally, the cut was sprayed with penicillin powder and then covered with aseptic dressing paste to prevent infection. The animals were returned to cages alone after surgery. At a specific time (3, 7, 14, and

21 days), the rats were sacrificed and the wounds along with surrounding tissues were collected. The collected tissues were fixed with 4% paraformaldehyde. After formaldehyde fixation for 1 week, the calcium was decalcified with 10% EDTA reagent for 2–4 weeks. Subsequently, H&E staining and toluidine blue staining were performed, then a fluorescence inverted microscope for microscopic was used examination and image acquisition analysis.

**Statistical analysis**. All data are shown as the mean ± standard deviation (SD). One-way analysis of variance (ANOVA) was applied for statistical analysis. $*p < 0.05$ and $**p < 0.01$ are considered significant and greatly significant difference, respectively.

**Reporting summary**. Further information on research design is available in the Nature Research Reporting Summary linked to this article.

## Data availability

Source data are provided with this paper. All experimental data within the article and its Supplementary Information are available from the corresponding authors upon request. The source data of Figs. 2, 3c, d, 5b, c, 6b, c, 7e, f and 9c, d and Supplementary Figs. 3, 4a, b, 6o, 7c, 8a, b, 13, 15b, c, and 16b, c, and Supplementary Tables 1, 2, and 3 are provided as a Source Data file. Source data are provided with this paper.

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

## Acknowledgements

This work is supported by Natural Science Foundation of China (22175037, 52103108, 21673093, and U1805234), Social Development of Instructive Program of Fujian Province (2020Y0020), Key Project for Advancing Science and Technology of Fujian Province ([2021]415-2021G02005), Natural Science Foundation of Fujian Province (2020J01147), Minjiang Scholar and startup fund for high-level talent at Fujian Normal University for Prof. W. Chen. We thank Prof. Ran Li of Fujian Agriculture and Forestry University, and Prof. Lixin Wu of Fujian Institute of Research on the Structure of Matter for plasma treatment of cotton gauze, Dr. Shuangquan Wu of Zhongnan Hospital of Wuhan University for generously providing Surgicel®.

## Author contributions

H. Liu conceived and directed the project. H. He, W. Zhou, A. Liu, W. Zhang, Y. Fang, and Y. Weng designed and performed the synthesis, characterization and hemostatic measurements of gauzes. F. Wang, S. Wang, and Z. Wang carried out the tests of biocompatibility. J. Gao, Y. Gao, and W. Chen run the density functional theory calculation and analysis. H. Liu and W. Chen wrote the manuscript.

## Competing interests

The authors declare no competing interests.

**Additional information**

