## [Peer Review File · Nature Communications]

A super-efficient and biosafe hemostatic cotton gauze with controlled balance of hydrophilicity/hydrophobicity and tissue adhesivenessReviewers' Comments:

Reviewer #1 (Remarks to the Author):

The authors presented a study on various hemostatic patches to determine hemostatic speed and amount of blood loss. They provided favourable evidence for the USO-g modified patch, which shows the least blood seepage through the patch. Can I ask the authors to address some concerns below?

1. It seems to be that the reported good results are simply due to an adjustment of the hydrophobicity of the woven patch, and nothing else. With the hydrophilic plain cotton, wetting is too fast, leading to blood loss, and with hydrophobic HTMS, wetting is too slow to provide a good seal at the seam between the patch and the tissue surface. With the USO-g, wetting is just at the right speed, to concurrently have the seal and the minimized blood loss.

The authors argued that the catechol plays an important role in giving the good results by providing tissue adhesion. They oxidized the catechol group and chelated it with iron to show that without the catechol group, blood loss increased. However, these experiments basically altered the hydrophobicity of the surface, and the altered results may be due to hydrophobicity changes, and not due to tissue adhesion effects. The claim of the catechol groups providing adhesion to tissue is not validated, and we are not sure whether it will attach to tissue or just to blood components. So, I think the authors may need additional work to show direct evidence of functionality beyond mere hydrophobicity adjustments.

In terms of hydrophobicity adjustment on hemostatic patches, in fact, commercial household bandages with "pain-free removal" already used this strategy. They placed a hydrophobic membrane in front of the cotton gauze in contact with the wound, and have micro-pores in the membrane, such that the surface touching the wound is less hydrophilic, and absorbs blood to a slower extent. The authors could buy some of these and scan them under the SEM to see these micro-pores. In view of these, the novelty of the study may need to be strengthened.

2. From video S5, it can be seen that the USO-g patch sticks to the wound surface substantially during removal, making the authors' claim that of no re-opening of wounds during patch removal somewhat surprising. The catechol groups were supposed to be more adhesive, potentially making it harder to remove. It is not clear if the hydrophobic groups play a role in reducing adhesion to the clot. It seems more likely that by reducing the blood stain area, removal becomes easier due to lower clot contact area, but the patch should still have high adhesion (force per area) to the clot and wet skin, and pulling it away should give similar risk of wound reopening. To make this claim of easy removal, I think the authors need additional verification and justification. Further, is the patch easier to remove compared to currently available commercial "pain-free / easy-removal" bandages?

3. In real life, the use of a bandage to reach hemostasis usually involves some pressure to press the bandage to the wound. With this pressure, the HTMS patch could perform better than USO-g patch in reducing blood loss. Can this be verified or refuted?

4. The claim that the USO-g patch enhance tissue growth is not yet credible. The only evidence is a thicker basal layer, which could already have been there from the beginning, or could be caused by other mechanism such as dehydration of the wound site by plain cotton patch. There is no direct evidence provided for tissue growth, and this claim is too important to be treated casually. In terms of the observation of inflammation level, microscopy images can also sometimes be influenced by white-balance settings or stain fading, and additional validation will be helpful. Further, the authors stated that less inflammation is found at the wound site, but inflammation is in fact beneficial towards wound healing. So it is not clear whether such an inflammatory difference will lead to enhanced or retard tissue growth.

5. Here are some claims which I feel that the authors do not yet have evidence for. The authors

should remove these claims or provide evidence.

a. The surface morphology of their patch is responsible for cell adhesion (page 7)
b. The USO-g patch causes coagulation that is independent of the body's physiological coagulation mechanism (pg 23). This one is actually difficult to agree with. The clotting is very likely to be due to both contact pathway and the tissue factor pathway, which are body's physiological mechanisms.

6. Since the authors agreed that it is difficult to keep taking the patch off to see if bleeding stopped, how was hemostasis time and bleeding time determined? Is the definition of hemostasis taken to be subjective in any way?

7. other minor issues

a. Sepsis is spelled wrong in the 1st paragraph

b. The abbreviations USO, ABO etc. should be explained the first time they are used, not at the end in the methods section.

c. Figure S10 (c) does not seem to be able to show cellular structures, and may be a result of imperfect staining

Reviewer #2 (Remarks to the Author):

The manuscript reports a new catechol-modified cotton gauze for hemostasis. The performance looks excellent so far, but the positive control groups were not suitable to evaluate the impact of this research. My comments to the authors are as follows.

(i) The number and/or surface density of modified ABO or USO should be characterized quantitatively. Discuss the relationship between surface modification and physical properties such as a dynamic contact angle.

(ii) Water absorption speed should be characterized by Byreck method.

(iii) Popular hemostat such as TachoSil and Surgicel should be used as control groups in the rat models. Otherwise, I can't evaluate if the present material is a superior material or not.

(iv) Ex vivo experiments using thromboelastography (TEG) such as ROTEM® (rotational thromboelastometry) or SonoClot are necessary to show the evidence of hemostatic mechanism based on the interaction between blood and the materials.

(v) In order to show the hypothesis in Figure 8, HE staining images and SEM images of the cross-section of the hemostat and damaged tissue after hemostasis is necessary.

(vi) The blood loss of QuickClot measured using the pig femoral artery model is desirable.

Point-to-point response to reviewers' comments

Reviewer #1 (Remarks to the Author):

The authors presented a study on various hemostatic patches to determine hemostatic speed and amount of blood loss. They provided favorable evidence for the USO-g modified patch, which shows the least blood seepage through the patch. Can I ask the authors to address some concerns below?

1. It seems to be that the reported good results are simply due to an adjustment of the hydrophobicity of the woven patch, and nothing else. With the hydrophilic plain cotton, wetting is too fast, leading to blood loss, and with hydrophobic HTMS, wetting is too slow to provide a good seal at the seam between the patch and the tissue surface. With the USO-g-gauze, wetting is just at the right speed, to concurrently have the seal and the minimized blood loss.

Author Reply: It is regretful that the reviewer reaches an initial impression that the good hemostatic performance of USO-g-gauze is simply due to an adjustment of the hydrophobicity of the woven patch, and nothing else. However, this is not the concepts which are conveyed by the results and analysis. Blood loss comes as a total of blood absorption in gauze and blood seepage at the seam of gauze/tissue surface. Therefore, we appreciate the opinion “With the USO-g-gauze, wetting is just at the right speed, to concurrently have the seal and the minimized blood loss”. Namely, for the hydrophilic standard cotton gauze, blood quickly diffuses freely in the gauze and weeps when blood loss exceeds the adsorption capacity of gauze patch. For the hydrophobic HTMS-g-gauze, blood doesn’t diffuse in the gauze, but severely seeps out of the seam at the gauze/tissue surface. While in the case of USO-g-gauze, blood slowly diffuses in controlled direction in the gauze, and less likely oozes out the seam at the gauze/tissue surface. In our manuscript, we tried to prove that the hydrophobic repellence and wet tissue adhesiveness from the special structure of USO (long alkyl chain terminated with a catechol group) play an essential role in this matter. Actually, adhesion to wet substrate is a big challenge for adhesive because interfacial water hinders adhesive from forming adhesive bonds with substrate (ZG Suo et al, Science, 2017, 357, 378; ZH Zhao et al, Nature, 2019, 575, 169). Catechol is a star group for adhesion to many sorts of wet substrates (GP Maier et al, Science, 2015, 349, 628; ZK Wang, Nat. Commun. 2017, 8, 2218). With the presence of catechol, USO-g-gauze may anchor to blood-wetted tissue to form dam-like barriers for hindering blood from seeping out of the seam at the gauze/tissue surface. While for its counterpart cotton gauze, this often happens when the gauze patch is saturated with blood.

The authors argued that the catechol plays an important role in giving the good results by providing tissue adhesion. They oxidized the catechol group and chelated it with iron to show that without the catechol group, blood loss increased. However, these experiments basically altered the hydrophobicity of the surface, and the altered results may be due to hydrophobicity changes, and not due to tissue adhesion effects.

Author Reply: It seems that the reviewer makes good arguments about hydrophobicity rather than tissue adhesion of gauze controls bleeding. But we'd like to prove that wet tissue adhesion is important for USO-g-gauze. First, let's compare HTMS-g-gauze and USO-g-gauze. In their structures, the feature in common is that both have long alkyl chains grafted on gauze surface, but the difference is that the latter's alkyl chain is terminated with a catechol group. The instant static WCA of HTMS-g-gauze and USO-g-gauze is 132.6° and 68.2°, respectively. They showed distinctly different hemostatic performance, namely blood doesn't diffuse in gauze but oozes out at the seam of HTMS-g-gauze/tissue surface, while it diffuses in gauze with no seepage at the seam of USO-g-gauze/tissue surface.

The instant static WCA of USOFe-g-gauze is 119° (higher than 68° of USO-g-gauze), but slowly reduces to 0° with diffusion of water droplet, similar to that occurred for USO-g-gauze (Fig. S12b). Given the similar wetting behavior of USO-g-gauze and USOFe-g-gauze, it is thought that the weak adhesiveness of catechol-Fe³⁺ groups to wet skin tissue is responsible for the substantially longer hemostatic time and more blood loss than USO-g-gauze and cotton gauze (p.23).

Fig. S12 (b) Water contact angle of USOFe-g-gauze with time.

The claim of the catechol groups providing adhesion to tissue is not validated, and we

are not sure whether it will attach to tissue or just to blood components. So, I think the authors may need additional work to show direct evidence of functionality beyond mere hydrophobicity adjustments.

Author Reply: Many thanks for suggesting measurement of tissue adhesion between gauze and tissue. In the revised manuscript, this adhesion was analyzed through theoretical calculation and instrumental measurement.

p.20-21. Firstly, from the viewpoint of interactions in the molecular level, density functional theory (DFT) calculations were performed to investigate the adsorption interaction of catechol group of USO-g-gauze with amino acid molecules, which are essential components of tissue keratin protein. The adsorption energies (ΔE_{ads}) of these amino acids to USO-g-gauze are calculated by considering the main non-covalent interaction modes including π - π stacking and hydrogen bond interactions.

The computed results reveal that the amino acids containing π -conjugated benzene ring, such as phenylalanine (F) and tyrosine (Y), can be effectively adsorbed on the catechol of USO-g-gauze through synergistic actions of π - π stacking and hydrogen bonding (Figs. 8C and S10), where the computed ΔE_{ads} values are as large as 0.621 and 0.729 eV, respectively. Comparatively, all the remaining fourteen amino acids without π -conjugated ring can effectively interact with the catechol of USO-g-gauze by adopting double hydrogen bonds, as presented in Fig. 8a and S10. All these hydrogen bonding distances are in the range of 1.69 ~ 1.80 Å (Table S1), and the calculated ΔE_{ads} values are as big as 0.570~ 0.639 eV (Fig. S10), indicating strong interaction force between them.

Furthermore, the effect of relative position between two OH groups on the benzene ring on ΔE_{ads} of a model amino acid glycine (G) on USO-g-gauze (Fig. S9) is examined. The computed results reveal that when the relative position between the two OH groups is changed from the original ortho- to meta- to para-arrangements, ΔE_{ads} values for G is reduced from 0.617 to 0.502/0.419 and then to 0.397/0.384 eV. Further, when the two OH groups are even separated by three H atoms, small ΔE_{ads} (*ca.* 0.383/0.355eV) is attained. Clearly, with increasing the

spacing distance between the two OH groups, ΔE_{ads} for the model amino acid decreases significantly, in view of the fact that double hydrogen bonds cannot be effectively formed or only a single hydrogen bond can be formed (Fig. S9). Therefore, the relative position between two OH groups on the benzene ring has an important influence on ΔE_{ads} , where the ortho position can bring the largest ΔE_{ads} due to formation of two hydrogen bonds. Further, when removing either of the two OH groups in catechol of USO-g-gauze (Fig. S10), the computed ΔE_{ads} values (0.248 and 0.355 eV) is about half of that of the corresponding structure with two hydrogen bonds, which means that both hydrogen bonds can be effectively formed simultaneously between the relevant amino acids and catechol in USO-g-gauze. Obviously, all of these can reflect the superior structural match to form double H-bonds between the amino acids and the catechol of USO.

Overall, USO-g-gauze with a long alkyl chain terminated with a catechol group can effectively interact with all these amino acids via double hydrogen bonds or the synergistic action of π - π stacking and hydrogen bonding. These non-covalent interactions contribute to USO-g-gauze's strong tissue adhesiveness, where the catechol at the end of alkyl chain can play a crucial role.

P.22, Secondly, apart from the above molecular level analyses of adhesion interaction, the adhesion force (or peeling force) of those gauzes on wet rat femoral tissue is measured and shows obvious variation from one to another. As expected, the hydrophobic HTMS-g-gauze has the lowest peeling force of 24 mN, while USO-g-gauze shows the largest force of 90 mN, which is ca. two times as much as that of cotton gauze (Fig. S11). From Suppl. Video 3, the adhesion of ABO-g-gauze and USO-g-chitosan to wound tissue is obviously perceivable when they were peeled off from the wounds, while it is less noticeable in the cases of cotton gauze etc (Suppl. Video 2). This vividly confirms the existence of adhesive interaction between catechol and tissue, but such non-covalent adhesion can be broken with mild peeling forces. In the case of the double -OH groups of catechol were modified such as with chelation with Fe^{3+} or oxidation into quinone, the tissue adhesion force

sharply decreases to values close to that of HTMS-g-gauze (Fig. S11). This further confirms that catechol group plays an essential role in the wet tissue adhesion.

Therefore, wetting, in another word, hydrophobicity/hydrophilicity of gauze is not the only requirement when our new gauze is designed. Adhesion to wet tissue is the other prerequisite for this gauze. A mere hydrophobicity control cannot make a good gauze like USO-g-gauze, as proved by USO-g-gauze and USOFe-g-gauze which have similar wetting property, but substantially different hemostatic time and blood loss (p.23).

In terms of hydrophobicity adjustment on hemostatic patches, in fact, commercial household bandages with “pain-free removal” already used this strategy. They placed a hydrophobic membrane in front of the cotton gauze in contact with the wound, and have micro-pores in the membrane, such that the surface touching the wound is less hydrophilic, and absorbs blood to a slower extent. The authors could buy some of these and scan them under the SEM to see these micro-pores. In view of these, the novelty of the study may need to be strengthened.

Author reply: According to the suggestion, SEM was used to observe the cross-sectional morphology of Bandage®. Its structure is just like the reviewer’s description, namely a hydrophobic layer is physically assembled on the surface of a cotton nonwoven. The main objective of this layer is anti-adhesion, in order to easily remove bandage without breaking the blood plug.

If the reviewer accepts our explanations that a combination of wetting and wet tissue adhesion contributes to the good hemostatic performance of USO-g-gauze, then it is quite clear that the concept and hemostatic mechanism of USO-g-gauze is new and quite different from that of Bandage. Additionally, the application scenarios of Bandage® and USO-g-gauze is quite different. The former is suitable for small bleeding wounds, while the latter can be used in both small and massive bleeding injuries, and in many types of wounds such as regular/irregular, shallow/deep/puncture, and compressible/non-compressible ones.

2. From video S5, it can be seen that the USO-g patch sticks to the wound surface substantially during removal, making the authors' claim that of no re-opening of wounds during patch removal somewhat surprising. The catechol groups were supposed to be more adhesive, potentially making it harder to remove. It is not clear if the hydrophobic groups play a role in reducing adhesion to the clot. It seems more likely that by reducing the blood stain area, removal becomes easier due to lower clot contact area, but the patch should still have high adhesion (force per area) to the clot and wet skin, and pulling it away should give similar risk of wound reopening. To make this claim of easy removal, I think the authors need additional verification and justification. Further, is the patch easier to remove compared to currently available commercial "pain-free / easy-removal" bandages?

Author reply: It should be pointed out that the noticeable sticking of USO-g-gauze to tissue is a proof for its adhesion to tissue.

We agree that the catechol group of USO-g-gauze makes it interact with blood clot and tissue more strongly than standard cotton gauze. Thus, no re-bleeding while peeling-off is really surprising. In fact, no secondary bleeding upon removal of USO-g-gauze is observed in the rat femoral artery injury and liver laceration models, and in the pig femoral artery injury model. It is also observed for USO-g-chitosan in the rat femoral artery injury. Therefore, this is a universal phenomenon for USO-g-gauze, rather than an occasional case.

#2 Reviewer recommended us to observe the blood-stained gauze after hemostasis. We think the SEM findings may help explain the no secondary bleeding for USO-g-gauze.

p. 24-26, It's well-known that erythrocytes are the key component of the primary blood clot. Thus, more erythrocytes are aggregated, bigger clot is formed, shorter bleeding time and less blood loss are attained. For USO-g-gauze, the blood fluid movement in gauze and around the gauze/tissue surface governed by the unique wetting property and tissue/cell adhesiveness facilitates aggregation of astonishingly massive

erythrocytes, as shown in Fig. 8B. There are so many congested erythrocytes that even fill the quadrilateral macro-holes among the warp and weft yarns of the first two gauze layers. The thickness of the erythrocyte layer accumulated on the 1st layer reaches as high as 220 μm . On one hand, such thick erythrocyte layers in the first two USO-g-gauze layers serve as clots for effectively controlling bleeding. On the other hand, the erythrocytes accumulated at the injury site is so enormous that removal of USO-g-gauze would take away part of erythrocytes, but some erythrocyte clots (Fig. 5d) are still on site to avoid re-bleeding.

However, the erythrocyte accumulation ability of cotton gauze is poor as suggested by its sparse distribution on yarns of the whole four cotton gauze layers (Fig. 8A), which is because erythrocytes move along with the flowing blood fluid to everywhere in the cotton patch, rather than group together to form a big/thick plug. With sparse distribution of erythrocytes on yarns, cotton gauze becomes an essential part of the blood plug (Fig. 8A). The plug would be easily broken since erythrocytes are removed along with peeling-off cotton gauze, leading to secondary bleeding.

3. In real life, the use of a bandage to reach hemostasis usually involves some pressure to press the bandage to the wound. With this pressure, the HTMS patch could perform better than USO-g patch in reducing blood loss. Can this be verified or refuted?

Author reply: For comparison, when HTMS-g-gauze is compressed onto a bleeding wound, blood spills and still seeps out even after 2-minute-compressing (Suppl. Video 3). This is due to the hydrophobic HTMS-g-gauze has a poor moisture management ability which is not helpful for hemostat. When applied to a bleeding trauma, the hydrophobic HTMS-g-gauze inhibits blood wetting, absorption, wicking and diffusion into the upper gauze layer (Fig. 3 and Table 1), and therefore poor hemostatic performance. (p.13-14)

4. The claim that the USO-g patch enhance tissue growth is not yet credible. The only evidence is a thicker basal layer, which could already have been there from the

beginning, or could be caused by other mechanism such as dehydration of the wound site by plain cotton patch. There is no direct evidence provided for tissue growth, and this claim is too important to be treated casually. In terms of the observation of inflammation level, microscopy images can also sometimes be influenced by white-balance settings or stain fading, and additional validation will be helpful. Further, the authors stated that less inflammation is found at the wound site, but inflammation is in fact beneficial towards wound healing. So it is not clear whether such an inflammatory difference will lead to enhanced or retard tissue growth.

Author reply: Thanks for your professional advices. This evaluation was carefully re-examined and new results and analysis were presented to replace the inaccurate information given in the original version. The main finding is USO-g-gauze causes no significant inflammatory responses, and shows same in vivo biocompatibility as cotton gauze.

p.29-30, Fig. 10 shows histological changes of the subcutaneous muscle tissue treated with cotton gauze and USO-g-gauze at specific time points (3, 7, 14, 21 days) using H&E staining and toluidine blue staining. In the cotton gauze-treated tissue, plenty of neutrophils arose around the gauze on the third day after implantation (Fig.10a) and the corresponding number were counted to be about 50 ± 5 (Fig. 10c). However, very few neutrophils were observed 7 days later, which reduced to about 2 ± 1 after 21 days. Meanwhile, USO-g-gauze had a similar inflammatory response. The density of neutrophils increased in the USO-g-gauze treated tissues since implantation, but rapidly decreased to 4 ± 1 after 7 days. In addition to neutrophils, mast cell was another critical effector of inflammation. As shown in the toluidine blue staining (Fig. 10b and d), several mast cells were observed in the tissue section contacting with the cotton gauze and the USO-g-gauze on the seventh days after implantation, but sharply reduced 14 days later within the cotton and the USO-g-gauze treated tissue. The subcutaneous implantation examinations reveal that the surface modification with USO did not compromise the biocompatibility of cotton gauze, and caused no significant inflammatory responses.

Fig.10 (a) H&E staining and (b) toluidine blue staining sections of subcutaneous tissue surrounding the gauzes at varied time intervals in the muscle implantation experiments using rats. The corresponding numbers of (c) neutrophil and (d) mast cells at different time intervals.

5. Here are some claims which I feel that the authors do not yet have evidence for. The authors should remove these claims or provide evidence.

a. The surface morphology of their patch is responsible for cell adhesion (page 7)

Author reply: Thanks for this suggestion. We deleted this claim in the revised manuscript.

b. The USO-g patch causes coagulation that is independent of the body's physiological coagulation mechanism (pg 23). This one is actually difficult to agree with. The clotting is very likely to be due to both contact pathway and the tissue factor pathway, which are body's physiological mechanisms.

Author reply: Thanks for this comment. We agree that both contact pathway and the tissue factor pathway actively play in the formation of blood clot when gauze is in contact with bleeding injuries. The hemostasis of USO-g-gauze is a physical effect rather than change of the body's normal physiologic clotting mechanism. Thus, we

delete this claim in the revised manuscript. (p.26)

6. Since the authors agreed that it is difficult to keep taking the patch off to see if bleeding stopped, how was hemostasis time and bleeding time determined? Is the definition of hemostasis taken to be subjective in any way?

Author reply: We define that hemostatic stage is reached when the blood-stained area on gauze doesn't expand and no blood oozes out from the seam at gauze/wound contact surface (p.35). In fact, in terms of bleeding control, blood loss is correlated well with bleeding time. So, we think that blood loss per unit time is a more accurate indicator of gauze's hemostatic efficiency than bleeding time, when the conditions for the hemostatic evaluation are kept same for each gauze. Thus, only blood loss (not hemostatic time) was presented in the pig injury models.

7. other minor issues

a. Sepsis is spelled wrong in the 1st paragraph

Author reply: Sorry for this spelling error. It was corrected in the revised manuscript.

b. The abbreviations USO, ABO etc. should be explained the first time they are used, not at the end in the methods section.

Author reply: Full names were given when they appeared the first time in the manuscript. (p.5)

c. Figure S10 (c) does not seem to be able to show cellular structures, and may be a result of imperfect staining

Author reply: To address this concern, in vivo inflammatory assay was re-evaluated. Experiment procedure (p.38-39), results (Fig. 10) and analysis (p.29-30) were updated in the revised manuscript.

Reviewer #2 (Remarks to the Author):

The manuscript reports a new catechol-modified cotton gauze for hemostasis. The performance looks excellent so far, but the positive control groups were not suitable to evaluate the impact of this research. My comments to the authors are as follows.

(i) The number and/or surface density of modified ABO or USO should be characterized quantitatively. Discuss the relationship between surface modification and physical properties such as a dynamic contact angle.

Author reply: We actually did preliminary experiments to optimize the synthesis formula and conditions. As listed in Table S1 (Section 1 of SI), four USO/ethanol solutions with USO concentration of 0.5, 1.0, 2.0 and 2.5% were used to make USO-g-gauze. The graft amount was ranged from 0.02 wt% to 0.28 wt% increasing with USO concentration. Water droplet instantly permeates and spreads in #1 and #2 gauzes, while it steadily stands on #4 within 60 s with WCA of ca. 120° (Fig. S1). #3 gauze has a unique wetting property, as detailed in our manuscript. In a pre-evaluation of their hemostatic ability in the rat femoral artery injury model, the hemostatic performance of #1 and #2 gauze is similar to that of standard cotton gauze, while #4 gauze is similar to HTMS-g-gauze. Therefore, #3 USO-g-gauze was prepared and used for the systematic study shown in the manuscript.

Table S1 WCA of a series of USO-g-gauze

USO-g-gauze	USO conc. in USO/ethanol solution (wt%)	Graft amount by weight (wt%)	WCA (°) at time of	
			0 s	60 s
#1	0.5	0.02	0	0
#2	1.0	0.04	0	0
#3	2.0	0.10	68.2 ± 2.1	0
#4	2.5	0.28	120.5 ± 3.5	120.2 ± 3.3

Fig. S1 WCA of #1, #2, #4 USO-g-gauze at 0 s and 60 s.

(ii) Water absorption speed should be characterized by Byreck method.

Author reply: Thanks for this nice suggestion. Moisture management was tested for these gauzes. Their results and analysis were presented in Table 1 and on p.9-10.

The movement of water in gauze was quantitatively measured by moisture management test (MMT). The variation of wetting time of gauzes (Table 1) is in good correlation to their wettability as shown in Fig. 3a and b. The wetting time (top side) is as short as 0.19 s for cotton and ABO-g-cotton gauzes, and substantially increases to 1.59 s for USO-g-cotton, and to 15.53 s for hydrophobic HTMS-g-gauze. As indicated in Table 1, the shorter wetting time is, the faster the water absorption rate and spreading speed is. The water absorption rate of gauzes follows a decreasing order of Cotton gauze > ABO-g-gauze > USO-g-gauze > HTMS-g-gauze, while the water spreading speed is in the order of Cotton gauze \approx ABO-g-gauze \gg USO-g-gauze \gg HTMS-g-gauze. The changing patterns of these indices suggest that the surface chemical structure of gauze effectively guides moisture movement (wetting, spreading, and diffusion) in gauze, agreeing well with the results shown in Fig. 3. The designed concomitant hydrophobic/hydrophilic structure of USO-g-cotton imparts it with not only proper wetting time and spreading rate, but its ability of water diffusion from one side to

the other side (indicated by the cumulative one-way transport capacity, Table 1). Such a unique property would be very helpful for controlling blood movement in gauze and at the gauze/tissue contact surface when it is practically applied as a topical hemostat, as will be shown in the following sections.

Table 1 Results of moisture management test of gauzes

Sample	Wetting time (s)		Maximum wetted radius (mm)		Water absorption rate (%/s)	Spreading rate (mm/s)	Cumulative one-way transport capacity
	Top side	Bottom side	Top side	Bottom side	Top side	Top side	
Cotton gauze	0.19±0.00	0.19±0.00	20.00±5.00	20.00±5.00	28.52±2.86	20.19±1.40	220.53±54.12
ABO-g-gauze	0.19±0.00	0.19±0.00	10.00±0.00	10.00±0.00	11.46±5.20	18.03±0.10	370.38±21.40
HTMS-g-gauze	15.53±0.58	48.65±5.65	5.00±0.00	3.33±0.36	3.06±0.17	0.320±0.01	843.65±56.37
USO-g-gauze	1.59±0.47	3.00±0.00	10.00±0.00	5.00±0.00	9.33±3.50	3.42±0.64	672.32±68.44

(iii) Popular hemostat such as TachoSil and Surgicel should be used as control groups in the rat models. Otherwise, I can't evaluate if the present material is a superior material or not.

Author reply: Application scenarios are different for TachoSil, Surgicel and cotton-type gauzes. A gauze is considered good in one scenario, but not good in another one.

Herein, we quote USFDA on the description of TachoSil

(<https://www.fda.gov/vaccines-blood-biologics/approved-blood-products/tachosil>):

“TachoSil is a **fibrin sealant patch** indicated for use with manual compression in adult and pediatric patients as an adjunct to hemostasis in cardiovascular and hepatic surgery, **when control of bleeding by standard surgical techniques (such as suture, ligature or cautery) is ineffective or impractical.**”

We quote Ethicon on the product information of SURGICEL® Absorbable Hemostat (<https://www.jnjmedicaldevices.com/en-US/product/surgicel-fibrillar-absorbable-hemostat>):

“It is used adjunctively in **surgical procedures to assist in the**

control of capillary, venous, and small arterial hemorrhage when ligation or other conventional methods of control are impractical or ineffective.” Therefore, Surgicel is often used in operation room for bleeding injuries where bio-absorbability is highly required; And it is not for the control of massive bleeding scenarios, and rarely used for First-Aid hemorrhaging control in emergent situations. Cotton gauze and QCG are common and standard gauzes for controlling external large bleeding area or massive bleeding wounds in battlefields and civilian accidents, and they are one of the essential First-Aid products. Thus, in the application scenario of controlling large bleeding area or massive bleeding wounds, standard cotton gauze and QCG rather than TachoSil/Surgicel are good candidate controls for our USO-g-gauze. The hemostatic results in rat and pig injury models have demonstrated USO-g-gauze’s much better performance over cotton gauze and QCG.

(iv) Ex vivo experiments using thromboelastography (TEG) such as ROTEM® (rotational thromboelastometry) or SonoClot are necessary to show the evidence of hemostatic mechanism based on the interaction between blood and the materials.

Author reply: According to the working mechanism of TEG, it is suitable for measuring the effect of medicine (in liquid or solution) on the blood coagulation dynamics and platelet activity. In our experience, we found that even a tiny fabric gauze in the cuvette with < 0.5 ml blood would affect the movement of the sensitive probe, and give unreliable signals.

For cotton gauze without any active additives, it quickly wicks blood fluid to concentrate red blood cells and platelets to form a primary clot, accompanied by coagulation cascade reactions to produce fibrin for tightening the clot. This physical hemostatic mechanism has been universally accepted.

For USO-g-gauze, wicking of blood fluid for aggregation of red blood cells and platelets still matters in the hemostasis, while repellence of blood fluid from hydrophobic alkyl chain as well as tissue adhesion also contributes to the hemostasis. In our response to the next concern you raised, SEM images clearly shows that very

thick erythrocyte layers (blood plug) are formed with the help of USO-g-gauze. In fact, all these effects belong to physical blocking of the bleeding injuries. In the meantime, body's physiological hemostatic mechanisms (contact pathway and the tissue factor pathway, as kindly suggested by #1 reviewer) take effect to yield thrombin for producing tough clot.

(v) In order to show the hypothesis in Figure 8, HE staining images and SEM images of the cross-section of the hemostat and damaged tissue after hemostasis is necessary.

Author reply: This is a great idea which we hope we had considered it when we conducted this research.

Method (p.35-36): For observing the micro-morphology of the blood-stained gauze layer after hemostasis, a patch stacked with four gauze layers was dressed onto the rat femoral artery injury to reach hemostasis. The gauze was placed in centrifuge tubes and fixed with 2.5% (v/v%) GA/PBS solution at room temperature for 4 h. Then it was rinsed twice with distilled water, followed by sequential dehydration in 0%, 10%, 20%, 30%, 40%, 50%, 60%, 70%, 80%, 90% and 100% (v/v%) ethanol/PBS solution. consequently, it was dried at 37 °C. The gauze was sputter-coated with gold before SEM observation.

Results: p.24-26, As shown in Fig. 8B, there are so many aggregated erythrocytes that they almost completely fill the quadrilateral macro-pores among warp and weft yarns of the first two gauze layers. The thickness of the erythrocyte layer accumulated on the 1st layer reaches as high as 220 μm . Such thick erythrocyte layers serve as clots for effectively controlling bleeding. The erythrocytes accumulated at the injury site is so enormous and thick that removal of USO-g-gauze would take away part of erythrocytes, but erythrocyte clots (Fig. 5d) are still on site to avoid re-bleeding.

However, the erythrocyte accumulation ability of cotton gauze is poor as suggested by its sparse distribution on cotton gauze yarns with none in the quadrilateral macro-pores of the whole four gauze layers (Fig. 8A). This is because erythrocytes move along with the fast blood wicking to everywhere in the cotton gauze patch, rather than group

together to form a big erythrocyte plug. With sparse aggregation of erythrocytes on yarns, cotton gauze becomes an essential part of the blood plug (Fig. 8A). The plug would be easily broken since erythrocytes are removed along with peeling-off cotton gauze, leading to secondary bleeding.

Fig. 8 Aggregation of erythrocytes on cotton gauze and USO-g-gauze after hemostasis in the rat femoral artery injury model; and hemostatic mechanism diagram of USO-g-

gauze. (a) On the cotton gauze patch composed of four stacked gauze layers, erythrocytes sparsely distribute on cotton yarns of all layers. The macro-pores among yarns are vacant. (b) On the USO-g-gauze patch composed of four stacked gauze layers, erythrocytes fill the macro-pores among yarns in the first two layers, but are absent in the pores and on the yarns of the 3rd layer. Cross-section of the 1st layer shows a thick erythrocyte layer. (c) Through adhesive bonds like π - π stacking interaction and hydrogen-bond between USO's catechol group and wound tissue's amino acid units, dam-like barriers form surrounding the wound. They retard blood oozing out from the tissue surface. The repelling pressure from the hydrophobic effect among long alkyl chains slows down blood wicking movement. (d) Dam-like barriers forming by USO between gauze layers and between fibers, retard blood diffusion in the vertical and horizontal directions, largely confine blood movement in the pores between warp and weft yarns, resulting accumulation of blood cells.

(vi) The blood loss of QuickClot measured using the pig femoral artery model is desirable.

Author reply: To address the reviewer's concern, QCG's blood loss in the pig femoral artery and skin laceration injury models was measured according to the same protocols as that for USO-g-gauze. The results were updated in the context on p.18-19, and in Fig. 7e and f.

In the pig femoral artery injury model, blood loss is 0.80 ± 0.12 g for USO-g-gauze, while that of cotton gauze, ABO-g-gauze, QCG, and HTMS-g-gauze is 5.12 ± 0.34 g, 4.16 ± 0.40 g, 3.93 ± 0.30 g, and 8.20 ± 0.34 g, respectively (Fig. 7e). Therefore, blood loss from the pig femoral artery wound treated with USO-g-gauze is only 15.6% and 20.4% of that with cotton gauze and QCG, respectively.

In the pig skin laceration model, blood loss of the cuts treated with cotton gauze, ABO-g-gauze, QCG, HTMS-g-gauze, and USO-g-gauze is 0.55 ± 0.04 g, 0.32 ± 0.03 g, 0.22 ± 0.02 g, 0.71 ± 0.02 g, and 0.032 ± 0.01 g (Fig. 7f), respectively. Compared with cotton gauze and QCG, the blood loss for USO-g-gauze reduces by ca. 94% and

85.5%, respectively.

Therefore, the results of the pig femoral artery injury and skin laceration models further justify that USO-g-gauze's hemostatic efficacy in severe bleeding wounds is superior to standard cotton gauze and QCG.

Fig. 7 Blood loss of gauzes in the pig (e) femoral artery and (f) skin injury models.

REVIEWER COMMENTS

Reviewer #1 (Remarks to the Author):

I thank the authors for their responses, and am satisfied that there's an adhesion mechanism on top of the adjusted hydrophilic absorption of blood.

The additional toxicity studies are also welcome, but it comes in potential conflict with toxicity information from the literature. According to the US CDC and EPA, there's literature showing that catechols cause contact dermatitis, and acute toxicity by oral or dermal exposure, among other hazards. Could the non-toxic responses found by the authors be due to small dosage? I would ask the authors to explain this to the readers.

Reviewer #2 (Remarks to the Author):

The data is interesting, but two points such as (i) fair evaluation of hemostatic function and (ii) the evidence of the the hypothesis of hemostatic mechanism are still concern.

I guess that the authors avoided the fair comparison with commercially available hemostats such as Tachosil. So far, surgeons rely on Tachosil best in the various hemostats. It is the most straightforward evidence of the superiority of this material. Animal experiments are the mimic of clinical situation in the end. Thus, it is desirable to compare the mechanism and fair hemostatic functions.

DFT calculation is interesting, but there is huge gap between functions and molecular level analysis since the structure of this material is complicated and dynamically change during hemostatic process by blood uptake. I guess that coated catechol would not show the acceleration of coagulation after adsorption of fibrinogen molecules on the surface of fibers. The comparison with Tahosil and Surgicel is not difficult.

Dear Dr. Robert Guillatt:

Many thanks for the referees' professional opinions on our manuscript entitled "A super-efficient and biosafe hemostatic cotton gauze with controlled balance of hydrophilicity/hydrophobicity and tissue adhesiveness (NCOMMS-20-05729A-Z)".

We have carefully revised our manuscript according to the great and insightful comments, and the formatting instructions of your journal. You may track the changes which were red-highlighted in the revised manuscript. A point-to-point response to reviewers' comments was listed at the end of this letter.

Sincerely Yours,

Prof. Dr. Haiqing Liu

Point-to-point response to reviewers' comments

Reviewer #1:

I thank the authors for their responses, and am satisfied that there's an adhesion mechanism on top of the adjusted hydrophilic absorption of blood.

The additional toxicity studies are also welcome, but it comes in potential conflict with toxicity information from the literature. According to the US CDC and EPA, there's literature showing that catechols cause contact dermatitis, and acute toxicity by oral or dermal exposure, among other hazards. Could the non-toxic responses found by the authors be due to small dosage? I would ask the authors to explain this to the readers.

Author reply: Thanks for pointing out this concern.

After surface immobilization onto gauze and fully cleaning off the free catechol compound USO, the USO-g-gauze is no harm to skin. We have played with

USO-g-gauze many times with bare hands, and have never caught dermatitis. In fact, polymerized USO has low cytotoxicity for dentistry application and has a long history as a safe coating for wooden bowl in oriental countries. A short explanation was given in p. 20 in the revised manuscript:

“It should be mentioned that the free small catechol compound USO may cause contact dermatitis according to the US CDC and EPA, but the USO immobilized on cotton gauze through surface grafting has shown no dermatitis. In fact, polymerized USO has low cytotoxicity for dentistry application and has a long history as a safe coating for wooden bowl in oriental countries.”

Reviewer #2 :

1. The data is interesting, but two points such as (i) fair evaluation of hemostatic function and (ii) the evidence of the hypothesis of hemostatic mechanism are still concern.

I guess that the authors avoided the fair comparison with commercially available hemostats such as Tachosil. So far, surgeons rely on Tachosil best in the various hemostats. It is the most straightforward evidence of the superiority of this material. Animal experiments are the mimic of clinical situation in the end. Thus, it is desirable to compare the mechanism and fair hemostatic functions.

Author reply: We feel really sorry that we did not make a comparison study between our gauze and the suggested gold standard gauzes in the first revision. In this second revision, we would be very happy to use both Tachosil and Surgicel as controls. However, the hemostatic efficacy of Surgicel was measured, while Tachosil was not tested due to availability issues in China market (We inquired hospitals and sales representatives in several regions in China, but none can provide Tachosil).

In consideration of the application scenario of Surgicel, non-compressible rat liver laceration injury model was adopted. We found: Although blood hardly diffused into the outmost layer, much blood seeped out of the seam of Surgicel/liver and stained the cotton gauze under the liver, resulting in relatively long hemostatic time and large blood loss. This finding proves that it is not ideal for controlling large

hemorrhaging, and it is consistent with the product description that it is used adjunctively in surgical procedures to assist in the control of capillary, venous, and small arterial hemorrhage. The new results were presented in the revised manuscript as follows:

p. 12, On this injury model, its hemostatic efficacy is also much superior to Surgicel®. Although blood hardly diffuses into the outmost layer, blood seeps out of the seam of Surgicel/liver and stains the cotton gauze under the liver, resulting in hemostatic time and blood loss of 198.5 ± 5.3 s and 0.47 ± 0.08 g, respectively (Supplementary Fig. 6 and Movie 5).

2. DFT calculation is interesting, but there is huge gap between functions and molecular level analysis since the structure of this material is complicated and dynamically change during hemostatic process by blood uptake.

Author reply: We do agree a gap exists between functions and molecular level analysis. Though the adhesion force between catechol and substrate surfaces has been quantitatively measured by many researchers (Proc. Natl. Acad. Sci. USA 2007, 104, 3782; J. Am. Chem. Soc. 2013, 135, 377) and this work as well, the theoretical calculation or simulation of this force has been regrettably missed so far. To fill this gap, this work demonstrates the force of non-covalent bonding interactions between catechol and amino acids by theoretical calculation for the first time. The DFT calculations show that the adhesion mainly originates from the formation of bidentate hydrogen bonds between the dihydroxy group of catechol and the amino acid molecules in the protein, or the synergy of the hydrogen bonding and the π - π interaction between the benzene rings. The DFT calculation model effectively unveils the essential reasons behind the large adhesion of gauze at the atomic level, and reveals that the catechol functional group with unique structure plays a decisive role in the large adhesion force of gauze to tissue. All the microscopic information obtained at the atomic level can provide the in-depth theoretical insight into the effective interaction between our gauze and skin tissue, and eventually help establish

a reasonable structure-function-mechanism connection of our gauze.

3. I guess that coated catechol would not show the acceleration of coagulation after adsorption of fibrinogen molecules on the surface of fibers.

Author reply: You are certainly right that the grafted catechol would not show acceleration of coagulation. In the initial stage, a big and soft primary blood clot made of aggregated red blood cells quickly forms to block the injury site, due to the balanced wettability and tissue adhesiveness of the gauze. In the following stage, fibrins would form by thrombin-lysis of fibrinogen molecules at a normal physiological speed, in order to reinforce the blood clot and get a mechanically hard thrombus.

A detailed discussion was given in P.17-19 in the revised manuscript.

4. The comparison with Tahosil and Surgicel is not difficult.

Author reply: We agree. The hemostatic performance of Surgicel on the rat liver laceration model was conducted. Results were shown in the revised manuscript.

REVIEWERS' COMMENTS

Reviewer #1 (Remarks to the Author):

I thank the authors for their responses, and I agree that this manuscript is ready for publication. Congratulations on a nice piece of work.

Reviewer #2 (Remarks to the Author):

The manuscript was well revised.

I hope that the materials will be compared to TacoSil in future since Tacosil showed better hemostatic effect than Surgicel due to fibrinogen coating.

I hope that the current material will be used in clinics.

Point-to-point response to reviewers' comments

Reviewer #1:

I thank the authors for their responses, and I agree that this manuscript is ready for publication. Congratulations on a nice piece of work.

Author response: Thank you for your compliments. We greatly appreciate your professional suggestions and comments in reviewing our manuscript.

Reviewer #2:

The manuscript was well revised.

I hope that the materials will be compared to TacoSil in future since Tacosil showed better hemostatic effect than Surgicel due to fibrinogen coating.

I hope that the current material will be used in clinics.

Author response: We really appreciate your professional suggestions and comments on our manuscript. We certainly will further work on this gauze and compare its hemostatic efficacy with other well-known gauzes including Tacosil. Currently we are working with partners to set up a company for producing and commercializing this gauze. We also hope it will be applied clinically for the well-being and life-saving of people.